# FedPSE: Personalized Sparsification with Element-wise Aggregation for Federated Learning

## Abstract

Federated learning (FL) is a popular distributed machine learning framework in which clients aggregate models' parameters instead of sharing their individual data. In FL, clients communicate with the server under limited network bandwidth frequently, which arises the communication challenge. To resolve this challenge, multiple compression methods have been proposed to reduce the transmitted parameters. However, these techniques show that the federated performance degrades significantly with Non-IID (non-identically independently distributed) datasets. To address this issue, we propose an effective method, called FedPSE, which solves the efficiency challenge of FL with heterogeneous data. FedPSE compresses the local updates on clients using Top-K sparsification and aggregates these updates on the server by element-wise average. Then clients download the personalized sparse updates from the server to update their individual local models. We then theoretically analyze the convergence of FedPSE under the non-convex setting. Moreover, extensive experiments on four benchmark tasks demonstrate that our FedPSE outperforms the state-of-the-art methods on Non-IID datasets in terms of both efficiency and accuracy.

## 1 Introduction

Federated learning (FL) is a prevailing distributed framework that can prevent sensitive data of clients from being disclosed (Kairouz et al., 2021; McMahan et al., 2017b). The naive FL includes three steps: uploading clients' models to the server after local training, global aggregation, and downloading the aggregated model from the server. In practice, weight updates $\Delta W = W_{new} - W_{old}$ can be communicated instead of model weights $W$ (Asad et al., 2021; Li et al., 2021a). Recently, FL is increasingly applied in multiple tasks, such as computer vision, recommender systems, and medical diagnosis (Bibikar et al., 2021; Kairouz et al., 2021; Qayyum et al., 2020; Xu et al., 2021).

### 1.1 Existing problem

Despite the aforementioned advantage, the communication cost of FL is overburdened by the fact that the server and clients exchange massive parameters frequently (Asad et al., 2021; Kairouz et al., 2021). Furthermore, there usually is a limited upstream/downstream bandwidth between the server and clients, such as wireless connection in the cross-device (ToC) FL and dedicated network in the cross-silo (ToB) setting, which further decreases the communication efficiency (Li et al., 2021a; Sattler et al., 2019). FL is much more time-consuming than traditional centralized machine learning, especially when the model parameters are massive under the cross-silo FL scenarios (Qayyum et al., 2020; Shi et al., 2020). Therefore, it is necessary to optimize the bidirectional communication cost to minimize the training time of FL (Bernstein et al., 2018; Philippenko & Dieuleveut, 2021; Sattler et al., 2019; Wen et al., 2017). In order to resolve the aforementioned challenge, various methods have been proposed, such as matrix decomposition (Li et al., 2021c; McMahan et al., 2017b), quantization (Li et al., 2021a; Sattler et al., 2019), and sparsification (Gao et al., 2021; Mostafa & Wang, 2019; Wu et al., 2020; Yang et al., 2021b). Although these novel algorithms can reduce the quantity of communicated information significantly, most of them can only work well

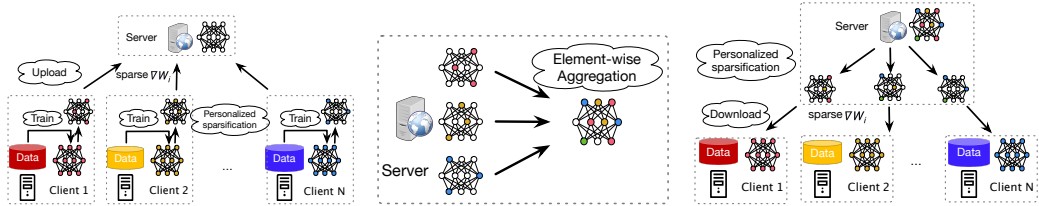

(a) Upstream personalized sparsification

(b) Element-wise aggregation

(c) Downstream personalized sparsification

Figure 1: The proposed framework of FedPSE.

under the ideal condition with IID (identically and independently distributed) datasets (Li et al., 2021c; Sattler et al., 2019; Wen et al., 2017). In fact, the isolated datasets in clients are usually heterogeneous, due to the reason that each dataset belongs to a particular client with a specific geographic location and time window of data collection Kairouz et al. (2021); Kulkarni et al. (2020); Xu & Huang (2022); Yang et al. (2021a). Hence, the current compression techniques, ignoring the personalization of clients, face a significant performance degradation on Non-IID datasets Liu et al. (2022); Sattler et al. (2019); Wu et al. (2020).

## 1.2 SOLUTION

To bridge this gap, we propose a Personalized Sparsification with Element-wise aggregation for the cross-silo federated learning (FedPSE) paradigm, as shown in Figure 1. For the first step of FedPSE, under the concern of efficiency and personalization, clients train their models with local datasets and upload the sparse updates to the server, as shown in Figure 1(a). The kept indices of these compressed updates are probably different from each other due to the heterogeneity of clients' datasets. Secondly, we leverage element-wise averaging to aggregate the collected sparse updates on the server, which can relieve the bias of the traditional aggregation method, as shown in Figure 1(b). Lastly, the server sparsifies the downstream parameters for each client in a personalized manner, as shown in Figure 1(c). Especially, the downstream updates, transferred from the server to each client, also possess individual $k$ elements to keep the overall compression ratio. Please see Section 4 for more details. To this end, FedPSE compresses both upstream and downstream communication overhead with personalization concerns.

## 1.3 CONTRIBUTION

We summarize our main contributions as follows:

- We propose a novel personalized sparsification with an element-wise aggregation framework for FL, which resolves the bidirectional communication challenge on Non-IID datasets.

- We propose an element-wise aggregation method, which can promote the performance of FL with sparse aggregated matrices.

- We propose a downstream selection mechanism to personalize the clients' models, which adapts to various distributions and significantly increases the performance in the Non-IID setting.

- We provide a convergence analysis of our method as well as extensive experiments on four benchmark datasets, and the results demonstrate that our proposed FedPSE outperforms the existing state-of-the-art FL framework on Non-IID datasets in terms of both efficiency and accuracy.

## 2 RELATED WORK

In this section, we briefly review optimization methods that focus on the core challenges in FL.

## 2.1 COMMUNICATION EFFICIENCY

Although FedAVG (McMahan et al., 2017a), the naive federated algorithm, can decrease the communication cost by allowing multiple local steps, the massive transmitted parameters in one communication step are still a critical bottleneck. In general, there are three kinds of compression

approaches to tackle this problem, i.e. matrix decomposition (Huang et al., 2022; Li et al., 2021c), quantization (Bernstein et al., 2018; Li et al., 2021a), and sparsification (Gao et al., 2021; Sattler et al., 2019; Wu et al., 2020).

Firstly, the decomposition method, decomposing the transmitted matrix, is unpractical with a smaller compression ratio and more computation complexity. Secondly, the quantization-based methods, limiting the number of bits, have the upper bound of the compression ratio and slow down the convergence speed in terms of training iterations (Chai et al., 2020). Thirdly, the sparsification-based methods mask some elements of the transmitted matrices, among which the Top-K sparsification with residual error is most widely used due to its promising performance and convergence guarantee (Dan Alistarh et al., 2018; Gao et al., 2021; Wu et al., 2020; Yang et al., 2021b). Therefore we leverage the error-compensated Top-K compressor to reduce the communication overhead in this article.

## 2.2 PERSONALIZATION

Since FL relies on the stochastic gradient descent (SGD) algorithm to train neural networks, the performance of FL is easily biased by the Non-IID datasets, e.g. feature skew and label skew, etc (Kairouz et al., 2021; Zhu et al., 2021). Different strategies have been applied to FL with heterogeneous concerns (e.g. data shuffling, multi-task learning, and local model optimization) (Achituve et al., 2021; Chen & Chao, 2021; Huang et al., 2021; Li et al., 2021b; Ma et al., 2022; Oh et al., 2021; T Dinh et al., 2020; Xu & Huang, 2022; Zhang et al., 2021). Although these state-of-the-art methods perform well under the Non-IID condition, the communication efficiency is often ignored.

## 2.3 COMPRESSION WITH PERSONALIZATION

In a word, few algorithms take both of the above two challenges into consideration. To our best knowledge, FedSTC (Sattler et al., 2019) and FedSCR (Wu et al., 2020) declare that they resolve the communication overburden with Non-IID data in one shot. Specifically, FedSTC compresses the clients' updates by combining Top-K sparsification and ternary quantization. Similarly, Fed-SCR compresses the transferred information by removing the redundant updates which are less than the adaptive threshold. However, the clients in FedSTC and FedSCR share the same model, which decreases the federated performance significantly in the Non-IID setting. In this paper, we propose a personalized compression paradigm for FL, which has competitive efficiency and promoted performance for heterogeneous data.

## 3 PRELIMINARY

In this section, we present some preliminary techniques of our proposal, including FedAVG and Top-K sparsification.

## 3.1 FEDAVG

FedAVG (McMahan et al., 2017a), a basic FL algorithm, builds distributed machine learning models via model aggregation rather than data aggregation. We suppose that there are $N$ clients with their datasets $\{D_1, D_2, ..., D_N\}$ in FL. Client $i, i \in \{1, \ldots, N\}$, trains the local model $W_i^{r-1}$ using $n_i$ samples individually and uploads the weight updates $\Delta W_i^r$ to the server for aggregation during the $r$-th federated round. Then the server leverage the averaging function to generate the global matrix: $\Delta W_s^r \leftarrow \sum_{i=1}^{N} \frac{n_i}{\sum_{i=1}^{N} n_i} \Delta W_i^r$. In the end, the global matrix $\Delta W_s^r$ is sent back to each client to update the local model $W_i^r \leftarrow W_i^{r-1} + \Delta W_s^r$.

## 3.2 TOP-K SPARSIFICATION

Top-K Sparsification is the most widely used compression method in FL, which remains a stable performance even with a high proportion of ignored parameters (Gao et al., 2021; Wu et al., 2020; Sattler et al., 2019; Yang et al., 2021b). The Top-K compressor selects K elements of the input matrix with the largest absolute values.

# 4 METHOD

In this section, we first give an overview of the proposed FedPSE paradigm. We then present its three main components, i.e., Upstream Personalized Sparsification (UPS), Element-Wise Aggregation (EWA), and Downstream Personalized Sparsification (DPS). Finally, we summarize the whole algorithm.

## 4.1 OVERVIEW

Our purpose of FedPSE is to design an FL paradigm that reduces the *bidirectional* communication cost while improving the performance on Non-IID datasets. We optimize the steps of FL according to its training process, as shown in Figure 1. Firstly, in order to reduce the upstream communication burden, we should *compress the information* from clients to the server. Secondly, as the global updates may be biased due to the compression, it is worth developing a *new aggregation method* that is suitable for sparse matrices. Thirdly, since clients with Non-IID datasets should have individual local models, the server needs to *personalize the downstream information* to each client while keeping the compression sparsity. Then we propose the motivation of our methods in the following paragraphs.

The first step, upstream personalized sparsification, compresses clients' updates after local training, which is inspired by the current work (Sattler et al., 2019; Wu et al., 2020). According to the prior research Wu et al. (2020), the matrices of weights updates are structure-sparse, in which most elements are redundant in the Non-IID setting. As a result, we leverage the Top-K sparsification operator with residual error to compress the weight updates, whose convergence rate has been theoretically proved (Gao et al., 2021; Haddadpour et al., 2021). Apparently, the reserved indices of compressed updates have its personalized distributions on heterogeneous datasets. Then each client transmits its individual updates to the server for the following aggregation.

---

**Algorithm 1: UPS**

**Input:** downstream personalized sparse updates $\Delta \hat{W}_{s,i}^{r-1}$, sparsity ratio $p$

**Output:** upstream sparse updates $\Delta \hat{W}_i^r$, samples number $n_i^r$

1   $W_i^{r,0} = W_i^{r-1,0} + \Delta \hat{W}_{s,i}^{r-1}$

2   Reset $n_i^r = 0$

3   **for** *each local training step* $t = 1, \ldots, T$ **do**

4      Sample a batch: $B_i^{r,t} \sim \mathcal{D}_i$

5      $g_i^{r,t} = \triangledown f_i(W_i^{r,t-1}, B_i^{r,t})$

6      $W_i^{r,t} = \mathbf{Opt}(W_i^{r,t-1}, g_i^{r,t}, \alpha)$

7      $n_i^r = n_i^r + \left\| B_i^{r,t} \right\|$

8   **end**

9   $\Delta W_i^r = W_i^{r,T} - W_i^{r,0} + e_i^{r-1}$

10   $\Delta \hat{W}_i^r = \mathbf{Top\text{-}K}(\Delta W_i^r, p)$

11   $e_i^r = \Delta W_i^r - \Delta \hat{W}_i^r$

12   **return** $\Delta \hat{W}_i^r, n_i^r$

---

The second step is proposing an appropriate aggregation method for the sparse updates. We demonstrate the bias of FedAVG with two examples under IID/Non-IID settings in Appendix A, which is caused by the sparsification. Then the effectiveness of the EWA method is proved, which is more suitable for sparse aggregation. Please see more details in Appendix A.

Thirdly, the server should sparsify the downstream updates for each client under the concern of personalization. In order to quantitatively measure the divergence between the global distribution and the local distribution, we compress the global updates, possessing $k$ elements, and compute its correlation distances with clients' upstream updates. Apparently, the correlation distances are likely different from each other in the Non-IID setting. Finally, the personalized downstream updates are selected by the correlation distances. Further details can be found in Section 4.4.

## 4.2 UPSTREAM PERSONALIZED SPARSFICATION (UPS)

We suppose that there are $N$ clients with their own training datasets $\{\mathcal{D}_1, \mathcal{D}_2, ..., \mathcal{D}_N\}$ and a server participated in FL. At the beginning of FL, client $i$ ($\forall i \in \mathcal{P}, \mathcal{P} = \{1, 2, ..., N\}$) initializes its weights $W_i$ with the same parameters $W_0$ (Zhao et al., 2018). Algorithm 1 presents the process of local training on client $i$ during $r$-th federated round, where the inputs, i.e. weight updates $\Delta \hat{W}_{s,i}^{r-1}$ and sparsity ratio $p$, are transmitted from the server. The first step of local training is updating the client model with the received $\Delta \hat{W}_{s,i}^{r-1}$ (line 1). Secondly, client $i$ trains its individual model with $n_i^r$

samples (lines 3-8), in which $f_i$ indicates the loss function and **Opt** (e.g., SGD or Adam) indicates the model's optimizer with the learning rate $\alpha$.

After that, the client $i$ compresses the updates via the error-compensated Top-K compressor (lines 9-11). Finally, the sparse updates $\Delta \hat{W}_i^r$ and the number of local training samples $n_i^r$ are uploaded to the server for aggregation.

## 4.3 ELEMENT-WISE AGGREGATION (EWA)

From the perspective of the selection mechanism in federated learning, the kept non-zero value in $\Delta \hat{W}_i^r$ indicates that the corresponding client $i$ is selected by the server at this element location. Apparently, each element of the aggregated matrix has its individual participating samples. We introduce our Element-Wise Aggregation (EWA) method in Algorithm 2. Firstly, the server receives the sparse updates $\Delta \hat{W}_i^r$ from client $i$ during $r$-th federated round, $\forall i \in \mathcal{P}$. Then the server computes the indices of nonzero values $M_{agg,i}^r$ for $\Delta \hat{W}_i^r$, which is used to calculate the sum matrix $\mathcal{N}$ of training samples for each location (lines 1-5). Finally, the server aggregates the global updates $\Delta W_s^r$, in which $\oslash$ indicates the element-wise division operator (line 6).

---

**Algorithm 2: EWA**

**Input:** sparse updates $\Delta \hat{W}_i^r$, samples number $n_i^r$, $i \in \mathcal{P}$

**Output:** aggregated updates: $\Delta W_s^r$

1 **for** $i = 1, \ldots, N$ **do**
2     Computes the index matrices of client updates:
$$M_{agg,i}^r = Sign\left(\left|\Delta \hat{W}_i^r\right|\right)$$
3 **end**
4 Sum matrix of training samples:
5 $\mathcal{N} = \sum_{i=1}^{N} n_i^r \cdot M_{agg,i}^r$
6 $\Delta W_s^r = \left(\sum_{i=1}^{N} n_i^r \cdot \Delta \hat{W}_i^r\right) \oslash \mathcal{N}$
7 **return** $\Delta W_s^r$

---

## 4.4 DOWNSTREAM PERSONALIZED SPARSIFICATION (DPS)

This process can be separated into three sub-steps: measurement of the heterogeneity between the global distribution and local distribution, selection of the downstream indices, and computation of the downstream personalized updates.

First of all, during the $r$-th federated round, the server quantifies the heterogeneity of data distributions via the distance between $\Delta \hat{W}_s^r$ and $\Delta \hat{W}_i^r, \forall i \in \mathcal{P}$ (lines 1-4). The server flattens the updates followed by normalization (lines 2-3). Then we compute the distance via the cosine function, which can be replaced by the other correlation functions (line 4). Secondly, we get the index matrix $M_s^r$ of the global sparse updates $\Delta \hat{W}_s^r$ via the sign function. In similarity, the index matrix $M_i^r$ of client $i$ is calculated (line 6). The intersection matrix $M_{i,in}^r$ of $M_s^r$ and $M_i^r$ can be regarded as the common information owned by both the server and client $i$, which will be wholly kept during downstream transmission (line 7). As $M_{i,in}^r$ has $k_{i,in}$ elements, we choose the rest $k - k_{i,in}$ elements from the compensation matrices $M_{i,c}^r$ and $M_{s,c}^r$. In order to enhance the generalization of training, we leverage the random mechanism to choose $d_{s,i}^r \cdot (k - k_{i,in})$ elements from $M_{i,c}^r$, while selecting $(1.0 - d_{s,i}^r) \cdot (k - k_{i,in})$ from $M_{s,c}^r$ (lines 8-13). Finally, we combine the personalized index matrix $M_{s,i}^r$ of client $i$ (line 14) and calculate the downstream sparse updates $\Delta \hat{W}_{s,i}^r$, from the server to client $i$ (lines 15-16).

Apparently, each client uploads its unique sparse updates $\Delta \hat{W}_i^r$, which means that the correlation distance $d_{s,i}^r$ and the downstream sparse updates $\Delta \hat{W}_{s,i}^r$ are different from each other.

## 4.5 PUTTING ALL TOGETHER

To sum up, we conclude the FedPSE framework in the Algorithm 4, which executes the federated process, named PSE, for R times. Before the training process, we initialize the model weights of clients as $W_0$, the received updates $\Delta W_{s,i}^0$ as zero and the sparsity ratio $p$ for communication compression concern. First of all, we get the personalized sparse updates $\Delta \hat{W}_i^r$ of each client $i$ using **UPS** method (Algorithm 1) with the last $\Delta \hat{W}_{s,i}^{r-1}$ and the sparsity ratio $p$ (line 5). Then we leverage the **EWA** operator (Algorithm 2) to compute the global updates ($\Delta W_s^r$) during $r$-th federated round

---

**Algorithm 3: DPS**

---

**Input:** Upstream sparse updates $\Delta \hat{W}_i^r$, global updates $\Delta W_s^r$, global sparse updates $\Delta \hat{W}_s^r$

**Output:** Downstream sparse updates $\Delta \hat{W}_{s,i}^r$ to the client $i$

1 # Measure the heterogeneity of distributions via $\Delta \hat{W}_s^r$ and $\Delta \hat{W}_i^r$

2 Normalize the global sparse updates: $\overline{\Delta} \hat{W}_s^r = Norm(Flatten(\Delta \hat{W}_s^r))$

3 Normalize the client $i$ sparse updates: $\overline{\Delta} \hat{W}_i^r = Norm(Flatten(\Delta \hat{W}_i^r))$

4 Compute the correlation distance: $d_{s,i}^r = 0.5 - 0.5 \cdot Cosine(\overline{\Delta} \hat{W}_s^r, \overline{\Delta} \hat{W}_i^r)$

5 # Select the reserved indices of downstream updates

6 Get the index matrices: $M_s^r = Sign\left(\left|\Delta \hat{W}_s^r\right|\right)$ and $M_i^r = Sign\left(\left|\Delta \hat{W}_i^r\right|\right)$

7 The intersection of index matrices: $M_{i,in}^r = M_i^r \odot M_s^r$

8 The number of non-zero value in $M_s^r$: $k = \|M_s^r\|$

9 The number of non-zero value in $M_{i,in}^r$: $k_{i,in} = \left\|M_{i,in}^r\right\|$

10 The compensations of index matrices: $M_{i,c}^r = M_i^r - M_{i,in}^r$ and $M_{s,c}^r = M_s^r - M_{i,in}^r$

11 Combine the indices:

12 $\widetilde{M}_{i,c}^r = Random(M_{i,c}^r, d_{s,i}^r \cdot (k - k_{i,in}))$

13 $\widetilde{M}_{s,c}^r = Random(M_{s,c}^r, (1 - d_{s,i}^r) \cdot (k - k_{i,in}))$

14 $M_{s,i}^r = M_{i,in}^r + \widetilde{M}_{i,c}^r + \widetilde{M}_{s,c}^r$

15 #Compute the downstream personalized updates: $\Delta \hat{W}_{s,i}^r = M_{s,i}^r \odot \Delta W_s^r$

16 **return** $\Delta \hat{W}_{s,i}^r$

---

(line 8). Furthermore, the server sparsifies the global updates $\Delta W_s^r$ and generates the $\Delta \hat{W}_s^r$ with the compression rate $p$ (line 9). Finally, we get the personalized updates of each client via **DPS** method (Algorithm 3) with the inputs of sparse upstream updates $\Delta \hat{W}_i^r$, global updates $\Delta W_s^r$ and sparse global updates $\Delta \hat{W}_s^r$ (line 11).

## 4.6 Theoretical analysis

In this subsection, we analyze the convergence results of the FedPSE framework theoretically. We suppose that the loss function ($f_i : \mathbb{R}^n \to \mathbb{R}$) of client $i$ ($\forall i \in \mathcal{P}$) is differentiable, where $n$ is the dimension of parameters. We consider the general setting in deep learning where $f_i$ is a non-convex function. Our convergence results are proved under the following assumptions:

**Assumption 1.** *Lipschitz Smoothness:*

*The loss function $f_i$ ($\forall i \in \mathcal{P}$) is L-Lipschitz smooth (L-smooth), i.e., $\|\nabla f_i(W_i^u) - \nabla f_i(W_i^v)\| \leq L\|W_i^u - W_i^v\|, \forall W_i^u, W_i^v \in \mathbb{R}^n$.*

**Assumption 2.** *Bounded Gradient:*

*The second moment of stochastic gradient $G_i$ calculated by a single sample within client $i$ is bounded, i.e., $\mathbb{E}[\|\sum_{i=1}^{N} G_i(W_i)\|^2] \leq \sigma^2, \forall W_i \in \mathbb{R}^n$.*

---

**Algorithm 4:** FedPSE framework

---

1 Initialization: $W_i^0 = W_0$,
    $\Delta W_{s,i}^0 = 0$ ($\forall i \in \mathcal{P}$), sparsity ratio $p$

2 **for** *each federated round $r = 1, ..., R$* **do**

3    # At clients:

4    **for** *each client $i \in \mathcal{P}$* **do**

5       $\Delta \hat{W}_i^r, n_i^r \leftarrow$ **UPS**($\Delta \hat{W}_{s,i}^{r-1}, p$)

6    **end**

7    # At Server:

8    $\Delta W_s^r \leftarrow$ **EWA**($\Delta W_i^r, n_i^r$), $\forall i \in \mathcal{P}$

9    $\Delta \hat{W}_s^r \leftarrow$ **Top-K**($\Delta W_s^r, p$)

10    **for** $i \in \mathcal{P}$ **do**

11       $\Delta \hat{W}_{s,i}^r \leftarrow$
    **DPS**($\Delta \hat{W}_i^r, \Delta W_s^r, \Delta \hat{W}_s^r$)

12    **end**

13 **end**

14 **return** $W_i^{R,T}, \forall i \in \mathcal{P}$

---

We aims to prove that $\min_{r \in \{1, \cdots, R\}} \mathbb{E}[\|\nabla f_i(W_i^r)\|^2] \overset{R \to \infty}{\longrightarrow} 0$, which is a normal convergence guarantee in the non-convex problem (Liu & Wright, 2015). Similar to Dan Alistarh et al. (2018),

we use $\tilde{W}_i^r$ to denote the auxiliary random variable during $r$-th federated round on client $i$, and $\tilde{W}^{r+1} = \tilde{W}^r - \alpha^r G(W^r)$, where $G(W^r) = \frac{1}{N}\sum_{i=1}^N G_i(W_i^r)$ and $\tilde{W}^0 = W_i^0 (\forall i \in \mathcal{P})$.

**Lemma 1.** *For any federated round* $r \geq 1$: $\mathbb{E}[||W_i^r - \tilde{W}_i^r||^2] \leq \frac{1+2\varphi}{\rho}\sum_{q=1}^r (\varphi(1+\rho))^q (\alpha^{r-q})^2 \frac{\sigma^2}{N}$, *where* $\varphi = 1 - \frac{k}{n}, 0 < k \leq n$ *and* $\rho > 0$.

We propose the bound of the difference between $\tilde{W}^r$ and $W_i^r$ during $r$-th federated round in Lemma 1. Consequently, the convergence of FedPSE is guaranteed as shown below.

**Theorem 1.** *Choose the learning rate schedule* $\alpha$, *s.t.* $\sum_{q=1}^r (\varphi(1+\rho))^q \frac{(\alpha^{r-q})^2}{\alpha^r} \leq C(\forall r > 0)$, *then our proposed FedPSE satisfies:*

$$\frac{1}{\sum_{r=1}^R \alpha^r}\sum_{r=1}^R \alpha^r \mathbb{E}[||\nabla f_i(W_i^r)||^2] \leq \frac{4(f_i(\tilde{W}^0) - f_i(\tilde{W}^*))}{\sum_{r=1}^R \alpha^r} + \frac{\frac{2\sigma^2 L}{N}(1 + \frac{2L(1+2\varphi)C}{\rho})\sum_{r=1}^R (\alpha^r)^2}{\sum_{r=1}^R \alpha^r}$$

*where constant* $C > 0$ *and* $\tilde{W}^*$ *denotes the optimal solution auxiliary variable.*

Theorem 1 implies that client model $W_i^r$ converges if federated round $R$ is large enough when $\alpha^r$ satisfies the following conditions: $\lim_{R\to\infty}\sum_{r=1}^R \alpha^r = \infty$, $\lim_{R\to\infty}\frac{\sum_{r=1}^R (\alpha^r)^2}{\sum_{r=1}^R \alpha^r} = 0$. Finally, we derive the convergence speed of our framework, please see more details and proofs in Appendix B.

## 5 EXPERIMENT

In this section, we empirically compare the performance of our proposed FedPSE framework with other federated learning paradigms for personalized compression. We aim to answer the following questions.

- **Q1:** whether FedPSE outperforms other optimized algorithms on the Non-IID data?
- **Q2:** whether the convergence speed of FedPSE is acceptable in the Non-IID setting?
- **Q3:** whether the correlation distance can be adopted to quantitatively measure the divergence between the local distribution and the global distribution?
- **Q4:** whether the **EWA** method can promote the performance of FedPSE?
- **Q5:** how does the sparsity ratio influence the performance of FedPSE?

### 5.1 EXPERIMENTAL SETTINGS

**Datasets.** To test the effectiveness of our proposed model, we choose four widely used benchmark datasets: MNIST (Deng, 2012), Fashion-MNIST (FMNIST) (Xiao et al., 2017), IMDB (Maas et al., 2011) and Cifar-10 (Krizhevsky et al., 2009). We load these datasets using the Keras package in Tensorflow2.8, keeping their original train/test samples (Abadi et al., 2016).

**Non-IID Setting.** As most empirical work on synthetic Non-IID datasets partitions a "flat" existing dataset based on the labels (Kairouz et al., 2021), we also use the label distribution skew as our Non-IID setting. Then we set a variable Non-IID ratio $\lambda$, ranging from 0.0 to 1.0, to simulate the Non-IIDness of clients' datasets, which is the same with Beutel et al. (2020). In this way, the distribution of clients' datasets is becoming more heterogeneous with a larger $\lambda$. Please see more details in Appendix C.1.

**Metrics.** Following the existing work (Sattler et al., 2019; Wu et al., 2020), we use accuracy on the test dataset as the evaluation metric. To compare the performance of different strategies in the decentralized scenario, we optimize their hyper-parameters and choose the average of metrics in all clients as the optimization target. Please see more details in Appendix C.2 and C.3.

### 5.2 PERFORMANCE COMPARISON OF ALGORITHMS WITH DIFFERENT NON-IID RATIOS

To answer the proposed question **Q1**, we firstly set different Non-IID ratios $\lambda$, i.e. 0.0, 0.5 and 1.0, to partition the original datasets for clients. As shown in Table 1 and Appendix D.1, we set the different

Table 1: Performance comparison of algorithms on datasets with different Non-IID ratios $\lambda$.

| Dataset | Clients | $\lambda$ | FedAVG | FedSTC | FedSCR | FedPSE | Dataset | FedAVG | FedSTC | FedSCR | FedPSE |
|---|---|---|---|---|---|---|---|---|---|---|---|
| MNIST | 2 | 1.0 | 0.9785 | 0.9927 | 0.9904 | **0.9967** | IMDB | 0.8406 | 0.8522 | 0.8961 | **0.9999** |
| | | 0.5 | 0.9901 | **0.9913** | 0.9851 | 0.9910 | | 0.8804 | 0.9010 | 0.8811 | **0.9013** |
| | | 0.0 | 0.9905 | **0.9915** | 0.9867 | 0.9910 | | 0.8810 | 0.8815 | 0.8815 | **0.8869** |
| | 5 | 1.0 | 0.9694 | 0.9907 | 0.9741 | **0.9939** | | 0.8494 | 0.9099 | 0.9375 | **0.9989** |
| | | 0.5 | 0.9899 | 0.9902 | 0.9816 | **0.9909** | | 0.8864 | 0.8941 | 0.8862 | **0.8977** |
| | | 0.0 | 0.9904 | **0.9910** | 0.9851 | 0.9908 | | 0.8827 | **0.8862** | 0.8841 | 0.8792 |
| | 10 | 1.0 | 0.9530 | 0.9876 | 0.9791 | **0.9940** | | 0.9080 | 0.9591 | 0.9633 | **0.9991** |
| | | 0.5 | 0.9725 | 0.9899 | 0.9801 | **0.9901** | | 0.8900 | 0.8922 | 0.8901 | **0.8951** |
| | | 0.0 | 0.9904 | **0.9905** | 0.9819 | 0.9865 | | **0.8908** | 0.8846 | 0.8770 | 0.8602 |
| FMNIST | 2 | 1.0 | 0.8781 | 0.8844 | 0.9031 | **0.9466** | Cifar10 | 0.7508 | 0.7611 | 0.7723 | **0.8576** |
| | | 0.5 | 0.8804 | 0.8926 | 0.8624 | **0.8953** | | 0.7712 | 0.7732 | 0.7731 | **0.7743** |
| | | 0.0 | **0.9003** | 0.8906 | 0.8523 | 0.8911 | | **0.7736** | 0.7711 | 0.7724 | 0.7721 |
| | 5 | 1.0 | 0.8068 | 0.8388 | 0.8403 | **0.9243** | | 0.7654 | 0.7786 | 0.7841 | **0.8602** |
| | | 0.5 | 0.8835 | 0.8952 | 0.8512 | **0.8967** | | 0.7760 | 0.7801 | 0.7822 | **0.7843** |
| | | 0.0 | **0.8995** | 0.8839 | 0.8641 | 0.8760 | | **0.7805** | 0.7805 | 0.7804 | 0.7780 |
| | 10 | 1.0 | 0.7909 | 0.8040 | 0.8305 | **0.9334** | | 0.7630 | 0.7721 | 0.7734 | **0.8741** |
| | | 0.5 | 0.8697 | 0.8850 | 0.8655 | **0.8910** | | 0.7800 | 0.7811 | 0.7832 | **0.7856** |
| | | 0.0 | **0.8993** | 0.8960 | 0.8644 | 0.8725 | | 0.7815 | **0.7822** | 0.7788 | 0.7770 |

numbers of clients, ranging from 2 to 100, for the cross-silo federated learning, which is similar to previous studies (Gao et al., 2021; Wu et al., 2020). Furthermore, we compare the accuracy of our proposed FedPSE and the three aforementioned algorithms, i.e. FedAVG McMahan et al. (2017a), FedSTC Sattler et al. (2019), and FedSCR (Wu et al., 2020). Among them, FedAVG is the baseline method without communication compression. In addition, FedSTC and FedSCR are state-of-the-art methods to solve the communication efficiency problem in the Non-IID setting. Especially, we set the same sparsity parameter $p = 0.9$ for FedSTC, FedSCR, and FedPSE, which means that only 10% parameters are transmitted between clients and the server compared with FedAVG. Then we summarize the results in Table 1 and Appendix D.1, which compare the performance of FedAVG, FedSTC, FedSCR, and FedPSE paradigms on these benchmark datasets.

From the results of the experiments, we conclude that our proposed FedPSE almost achieves the best performance of all algorithms in the Non-IID setting ($\lambda = 1.0 \ or \ 0.5$), which is also robust to the number of clients. For example, the average metric of FedPSE outperforms 6.85% for FedSTC and 5.57% for FedSCR on FMNIST dataset as shown in Table 2 of Appendix D.1. Besides, FedPSE is also robust to the Non-IID ratios, which can get very similar performance compared with FedAVG and FedSTC, while FedSCR has a weakness when the data distribution is more symmetrical ($\lambda = 0.0$). Furthermore, our method also outperforms other models with partial clients participation for aggregation as shown in Appendix D.2.

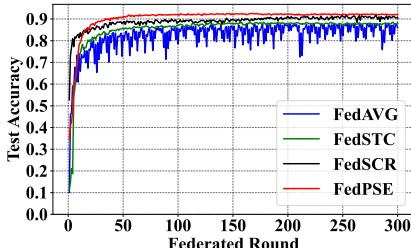

Figure 2: Test accuracy regarding rounds.

## 5.3 Convergence speed of FedPSE in the Non-IID setting

To answer the proposed question **Q2**, we plot the metric over the federated rounds on the Non-IID ($\lambda = 1.0$) FMNIST dataset as shown in Figure 2, in which the compression ratio $p$ is 0.9 with 5 participated clients. We observe that the performance of FedAVG is unstable on Non-IID datasets, which is also proposed in the previous result (Zhao et al., 2018), while other methods suffer the least from Non-IID data. It is essential that our proposed FedPSE, compared with FedSTC and FedSCR, improves the performance with a faster convergence speed. Furthermore, FedSCR performs best in the Non-IID setting, which is consistent with the former result (Wu et al., 2020).

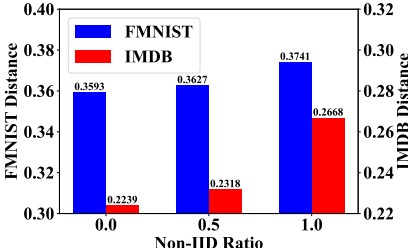

Figure 3: Correlation distance on 5 clients.

## 5.4 Correlation distance of FedPSE with different Non-IID ratios

To answer the proposed question **Q3**, we plot the correlation distance of participated clients with different Non-IID ratios (i.e. $\lambda = 0.0, 0.5, 1.0$) as shown in Figure 3.

Figure 3 shows that the average correlation distance increases over the Non-IID ratios, which demonstrates that the correlation distance is able to represent the heterogeneity between a local distribution and global distribution. Furthermore, we compute the variance of correlation distance under diverse Non-IID settings as shown in Appendix D.3. Apparently, the variance is becoming larger with the growth of heterogeneity, which means that the correlation distance also has a relationship with the distribution divergence among each client.

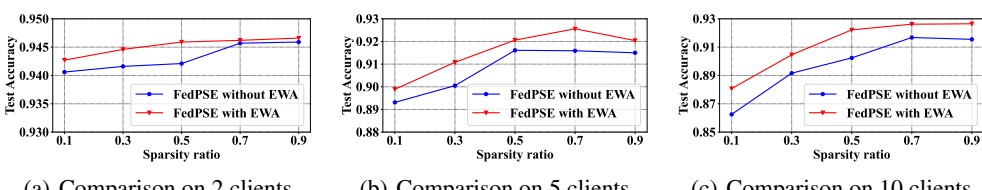

|  (a) Comparison on 2 clients | (b) Comparison on 5 clients | (c) Comparison on 10 clients |

Figure 4: Accuracy comparison of EWA in FedPSE over compression sparsity ratios on Non-IID datasets.

## 5.5 Performance comparison of FedPSE with different aggregation methods

To answer the proposed question **Q4**, we vary sparsity ratio $p$ from 0.1 to 0.9 on FMNIST dataset and report the average test accuracy on Non-IID datasets ($\lambda = 1.0$) in Figure 4, in which FedPSE without EWA indicates FedPSE using the naive aggregation method of FedAVG. Figure 4 shows that FedPSE with EWA consistently outperforms FedPSE without EWA under different compression rates, which demonstrates the effectiveness of our proposed EWA method. Besides, the average promotion of EWA is 0.002 on 2 clients, while the promotion is 0.0071 on 5 clients and 0.0143 on 10 clients. We can indicate that the promotion of EWA increases with the number of clients.

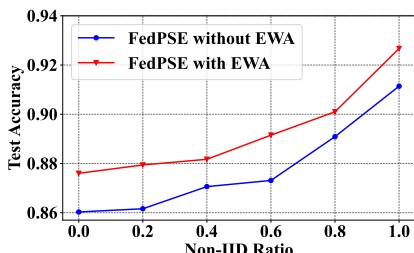

Figure 5: Accuracy with Non-IID ratios.

Furthermore, we construct experiments on the FMNIST datasets over Non-IID ratios $\lambda$ (i.e. from 0.0 to 1.0) with a fixed $p = 0.9$ as shown in Figure 5. The test accuracy of FedPSE with EWA exceeds the algorithm without EWA through different distributions. The performance of FedPSE increases rapidly with the rising $\lambda$, which means that our proposed paradigm performs better under the extreme Non-IID condition. In conclusion, the EWA method is useful for the sparse aggregation process in the server.

## 5.6 Performance comparison of FedPSE with different sparsity ratios

Figure 4 can also answer the proposed question **Q5**. We can find that the accuracy of FedPSE over sparsity ratios changes individually on the different number of clients. For instance, the accuracy keeps steady with most sparsity ratios on two clients, while the metric increases rapidly with the rising $p$ from 0.1 to 0.7 on more clients. We can optimize the sparsity hyper-parameter $p$ of FedPSE to balance the performance and efficiency of FedPSE.

## 6 Conclusion

We propose a personalized sparsification with element-wise aggregation for federated learning to solve the Non-IID isolated scenario. We first sparsify the upstream updates of clients via the Top-K operator. Then we propose the EWA aggregation method to promote the federated performance for sparse matrices. Finally, we leverage the DPS method to keep the personalization and sparsification for the downstream information. Experiments on real-world datasets demonstrate that our model significantly outperforms the current methods on the isolated Non-IID data.

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

## A    COMPARISON OF DIFFERENT AGGREGATION METHODS

In this section, we give two examples to compare the results of different aggregation methods under the IID and Non-IID settings.

For the IID setting, we suppose that there are three clients with their updates, i.e. $\Delta W_1 = (1,1)$, $\Delta W_2 = (2,2)$ and $\Delta W_3 = (3,3)$. It is evident that the original aggregation of these dense updates is $(2,2)$ (red arrow) as shown in Figure 6(a), which is regarded as the accurate result. We then generate the correspond-

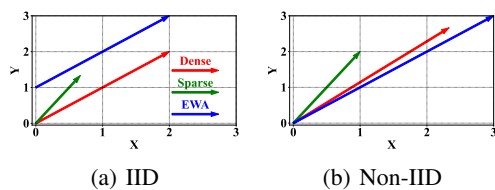

(a) IID     (b) Non-IID

Figure 6: Aggregation results of different methods.

ing sparse updates, i.e. $\Delta \hat{W}_1 = (0,1)$, $\Delta \hat{W}_2 = (2,0)$ and $\Delta \hat{W}_3 = (0,3)$, whose biased result $\left(\frac{2}{3}, \frac{4}{3}\right)$ (green arrow) is easily aggregated via the naive averaging method (McMahan et al., 2017b). From a microcosmic perspective, if we compute the aggregated vector via the Element-Wise Aggregation (EWA) method as shown in Section 4.3, we would get the precise aggregation $(2,2)$ (blue vector) with the above sparse updates.

In similarity, we assume that the clients' updates are $\Delta W_1 = (1,2)$, $\Delta W_2 = (3,2)$ and $\Delta W_3 = (3,4)$ in the Non-IID setting. Then we sparsify these updates via Top-K operator, i.e. $\Delta \hat{W}_1 = (0,2)$, $\Delta \hat{W}_2 = (3,0)$ and $\Delta \hat{W}_3 = (0,4)$. It's easy to calculate the accurate aggregation, $\left(\frac{7}{3}, \frac{8}{3}\right)$ (red arrow), the naive result, $(1,2)$ (green arrow), and the EWA vector, $(3,3)$ (blue arrow). Apparently, the EWA result is more close to the accurate dense aggregation than the naive sparse aggregation as shown in Figure 6(b), when it refers to the direction and magnitude.

Overall, the original averaging approach is easily biased by the sparse matrices, while our proposed EWA method is more suitable for the sparse aggregation.

## B    CONVERGENCE ANALYSIS

### B.1    ANALYSIS PRELIMINARIES AND ASSUMPTIONS

The framework FedPSE is used to minimize the differentiable loss function $f_i : \mathbb{R}^n \to \mathbb{R} \, (\forall i \in \mathcal{P})$ individually, where $n$ indicates the dimension of parameters. We consider the general setting in deep learning where $f_i$ is a non-convex function. Our convergence results are proved under the following assumptions:

**Assumption 3.** *Smoothness*: *The loss function $f_i \, (\forall i \in \mathcal{P})$ is L-Lipschitz smooth (L-smooth), i.e.,* $||\nabla f_i(W_i^u) - \nabla f_i(W_i^v)|| \le L||W_i^u - W_i^v||, \forall W_i^u, W_i^v \in \mathbb{R}^n.$

Then we assume that client $i$ trains its model $W_i$ with unbiased stochastic gradient, i.e., $E[G_i(W_i)] = \nabla f_i(W_i)$, in which $G_i$ denotes the stochastic gradient with a batch of $M$ samples.

**Assumption 4.** *Bounded Gradient*: *The second moment of local gradient $G_i$ is bounded, i.e.,* $\sum_{i=1}^{N} \mathbb{E}[||G_i(W_i)||^2] \le \sigma^2, \forall W_i \in \mathbb{R}^n$, *where $||.||$ is $\ell_2$-norm.*

As shown in Algorithm 5, FedPSE includes three steps, i.e., upstream personalized sparsification (UPS), element-wise aggregation (EWA), and downstream personalized sparsification (DPS). The training steps of each federated round are termed as *PSE* collectively in the following sections. The local model is updated during the $(r+1)$-th federated round by the following expression:

$$W_i^{r+1} = W_i^r - \text{PSE}_{i=1}^{N}(\alpha^r G_i(W_i^r) + e_i^{r-1}) \tag{1}$$

where $\alpha^r$ is the learning rate.

According to Algorithm 2, $e_i^{r-1} = \alpha^{r-1}G_i(W_i^{r-1}) - \text{Top-k}(\alpha^{r-1}G_i(W_i^{r-1}) + e_i^{r-2})$. We then expand the *PSE* with the previous parameters and get

$$
\begin{aligned}
&\text{PSE}_{i=1}^N(\alpha^r G_i(W_i^r) + e_i^{r-1}) \\
=&(M_{in}^r + \text{R}(M_{i,c}^r, d_{s,i}^r \cdot k_c) + \text{R}(M_{s,c}^r, (1-d_{s,i}^r) \cdot k_c) \odot \Delta W_s^r \\
=&(M_{in}^r + \widetilde{M_{i,c}^r} + \widetilde{M_{s,c}^r}) \odot \Delta W_s^r \\
=&W^r \odot (M_{in}^r + \widetilde{M_{i,c}^r}) \odot \text{Top-k}(\sum_{i=1}^N \text{Top-k}(\alpha^r G_i(W_i^r) + e_i^{r-1})) \\
&+ \widetilde{M_{s,c}^r} \odot \text{Top-k}(\alpha^r G_i(W_i^r) + e_i^{r-1})
\end{aligned}
\tag{2}
$$

where Top-k$(\cdot)$ is the compress operator which is described in Algorithm 1, R$(\cdot)$ denotes the random selection function in Algorithm 4 and $W^r = \vec{1}_n \oslash \sum_{i=1}^N \text{Sign}\left(\left|\text{Top-k}(\alpha^r G_i(W_i^r) + e_i^{r-1})\right|\right)$ is the element weight matrix.

Similar to Dan Alistarh et al. (2018), we use $\tilde{W}_i^r$ to denote the auxiliary model weight with convergence guarantee during $r$-th federated round on client $i$, and get

$$
\tilde{W}^{r+1} = \tilde{W}^r - \alpha^r G(W^r)
\tag{3}
$$

where $G(W^r) = \frac{1}{N}\sum_{i=1}^N G_i(W_i^r)$ and $\tilde{W}^0 = W_i^0(\forall i \in \mathcal{P})$. The difference between the auxiliary model $\tilde{W}^r$ and the local model $W_i^r$ can be represented by

$$
\begin{aligned}
&W_i^r - \tilde{W}^r \\
=&W_i^0 - \sum_{q=1}^r \text{PSE}_{i=1}^N(\alpha^q G_i(W_i^q) + e_i^{q-1}) - \tilde{W}^0 + \sum_{q=1}^r \alpha^q G(W^q) \\
=&\frac{1}{N}\sum_{q=1}^r\sum_{i=1}^N \alpha^q G_i(W_i^q) - \sum_{q=1}^r \text{PSE}_{i=1}^N(\alpha^q G_i(W_i^q) + e_i^{q-1})
\end{aligned}
\tag{4}
$$

Then we define a commonly used $k$-contraction operator (Gao et al., 2021; Sebastian U. Stich, 2018) as shown below:

**Definition 1.** *For any vector $W \in \mathbb{R}^n$ and $0 < k \le n$, operator $Q$ $(\mathbb{R}^n \to \mathbb{R}^n)$ is a $k$-contraction operator if it satisfies the following property:*

$$
\mathbb{E}||W - Q(W)||^2 \le (1 - \frac{k}{n})||W||^2, \forall W \in \mathbb{R}^n
\tag{5}
$$

Apparently, the Top-K compressor is a $k$-contraction operator.

**Lemma 2.** $\forall W_i \in \mathbb{R}^n$ *and* $0 < k \le n$, *we have*

$$
\begin{aligned}
&\mathbb{E}[||\frac{1}{N}\sum_{i=1}^N(\alpha^r G_i(W_i^r) + e_i^r) - PSE_{i=1}^N(\alpha^r G_i(W_i^r) + e_i^r)||^2] \\
\le&(1 - \frac{k}{n})||\frac{1}{N}\sum_{i=1}^N(\alpha^r G_i(W_i^r) + e_i^r)||^2
\end{aligned}
\tag{6}
$$

*Proof.* Combining equation 2 and Definition 1, we obtain

$$
\begin{aligned}
&\mathbb{E}[||\frac{1}{N}\sum_{i=1}^N(\alpha^r G_i(W_i^r) + e_i^r) - \text{Top-k}_{i=1}^N(\alpha^r G_i(W_i^r) + e_i^r)||^2] \\
\le&\mathbb{E}[||\frac{1}{N}\sum_{i=1}^N(\alpha^r G_i(W_i^r) + e_i^r) - \text{PSE}_{i=1}^N(\alpha^r G_i(W_i^r) + e_i^r)||^2] \\
\le&\mathbb{E}[\frac{1}{N}\sum_{i=1}^N(\alpha^r G_i(W_i^r) + e_i^r) - \text{Q}(\mathbb{E}[||\frac{1}{N}\sum_{i=1}^N(\alpha^r G_i(W_i^r) + e_i^r)]) \\
\le&(1 - \frac{k}{n})||\frac{1}{N}\sum_{i=1}^N(\alpha^r G_i(W_i^r) + e_i^r)||^2
\end{aligned}
$$

$\square$

## B.2  MAIN PROCESS

As a standard situation in non-convex settings (Liu & Wright, 2015), we want to guarantee the following convergence:

$$\min_{r \in \{1, \cdots, R\}} \mathbb{E}[||\nabla f_i(W_i^r)||^2] \xrightarrow{R \to \infty} 0$$

that is, the algorithm converges ergodically to the optimal point where gradients are zero. Our purpose is to minimize the difference between the "real" model $\tilde{W}_i^r$ and the viewed $W_i^r$ observed at federated round $r$, which means decreasing the value of loss function $f_i(W_i^r)$. Furthermore, we need to bound the following expression:

$$\frac{1}{\sum_{r=1}^{R} \alpha^r} \sum_{r=1}^{R} \alpha^r \mathbb{E}[||\nabla f_i(W_i^r)||^2]$$

Then We get Lemma 3:

**Lemma 3.** *For any federated round $r \geq 1$:*

$$\mathbb{E}[||W_i^r - \tilde{W}_i^r||^2] \leq \frac{1 + 2\varphi}{\rho} \sum_{q=1}^{r} (\varphi(1+\rho))^q (\alpha^{r-q})^2 \frac{\sigma^2}{N} \tag{7}$$

*where $\varphi = 1 - \frac{k}{n}, 0 < k \leq n$ and $\rho > 0$.*

*Proof.* We derive the difference between $W_i^{r+1}$ and $\tilde{W}^{r+1}$:

$$\mathbb{E}[||W_i^{r+1} - \tilde{W}_i^{r+1}||^2]$$

$$= \mathbb{E}[||W_i^r - \text{PSE}_{i=1}^N(\alpha^r G_i(W_i^r) + e_i^r) - \tilde{W}^r + \alpha^r G(W^r)||^2]$$

$$= \mathbb{E}[||W_i^r - \tilde{W}^r - \text{PSE}_{i=1}^N(\alpha^r G_i(W_i^r) + e_i^r) + \frac{\alpha^r}{N} \sum_{i=1}^{N} G_i(W_i^r)||^2]$$

$$= \mathbb{E}[||W_i^r - \tilde{W}^r - \frac{1}{N} \sum_{i=1}^{N} e_i^r - \text{PSE}_{i=1}^N(\alpha^r G_i(W_i^r) + e_i^r) + \frac{1}{N} \sum_{i=1}^{N} (\alpha^r G_i(W_i^r) + e_i^r)||^2]$$

$$\leq \mathbb{E}[||\frac{1}{N} \sum_{i=1}^{N} (\alpha^r G_i(W_i^r) + e_i^r) - \text{PSE}_{i=1}^N(\alpha^r G_i(W_i^r) + e_i^r)||^2] + \mathbb{E}[||W_i^r - \tilde{W}^r - \frac{1}{N} \sum_{i=1}^{N} e_i^r||^2]$$

From Lemma 2, we can obtain

$$\mathbb{E}[||W_i^{r+1} - \tilde{W}_i^{r+1}||^2]$$

$$\leq \varphi ||\frac{1}{N} \sum_{i=1}^{N} (\alpha^r G_i(W_i^r) + e_i^r)||^2 + \mathbb{E}[||W_i^r - \tilde{W}^r - \frac{1}{N} \sum_{i=1}^{N} e_i^r||^2]$$

$$\leq \varphi ||\frac{1}{N} \sum_{i=1}^{N} \alpha^r G_i(W_i^r)||^2 + \varphi ||\frac{1}{N} \sum_{i=1}^{N} e_i^r||^2 + \mathbb{E}[||W_i^r - \tilde{W}^r - \frac{1}{N} \sum_{i=1}^{N} e_i^r||^2]$$

From equation (4) and Lemma 2, we have

$$\mathbb{E}[||W_i^{r+1} - \tilde{W}_i^{r+1}||^2]$$

$$\leq \varphi \mathbb{E}||\alpha^r G(W^r)||^2 + \varphi \mathbb{E}||W_i^r - \tilde{W}^r||^2$$

$$+ (1+\varphi) \mathbb{E}||\frac{1}{N} \sum_{q=1}^{r} \sum_{i=1}^{N} \text{Top-k}(\alpha^r G_i(W_i^r) + e_i^r) - \sum_{q=1}^{r} \text{PSE}_{i=1}^N(\alpha^q G_i(W_i^q) + e_i^q)||^2$$

$$\leq \varphi \mathbb{E}||\alpha^r G(W^r)||^2 + \varphi \mathbb{E}||W_i^r - \tilde{W}^r||^2 + (1+\varphi) \mathbb{E}||W_i^{r+2} - W_i^{r+1}||^2$$

$$\leq (1+2\varphi)(1+\frac{1}{\rho}) \mathbb{E}||\alpha^r G(W^r)||^2 + \varphi(1+\rho) \mathbb{E}||W_i^r - \tilde{W}^r||^2$$

Iterate the above inequality by $r$ to get:

$$\mathbb{E}[||W_i^r - \tilde{W}_i^r||^2] \leq \frac{1 + 2\varphi}{\rho} \sum_{q=1}^{r} (\varphi(1+\rho))^q \mathbb{E}||\alpha^{r-q} G(W^{r-q})||^2$$

From Assumption 4 and Lemma 3, we have

$$\mathbb{E}[||W_i^r - \tilde{W}_i^r||^2] \leq \frac{1 + 2\varphi}{\rho} \sum_{q=1}^{r} (\varphi(1+\rho))^q \mathbb{E}||\alpha^{r-q} G(W^{r-q})||^2$$

$$\leq \frac{1 + 2\varphi}{\rho} \sum_{q=1}^{r} (\varphi(1+\rho))^q (\alpha^{r-q})^2 \frac{\sigma^2}{N}$$

$\square$

**Theorem 2.** *Assume that our proposed FedPSE is applied to minimize the objective loss function $f_i$ that satisfies the assumptions in A.1. If we choose a learning rate schedule that satisfies:*

$$\sum_{q=1}^{r} (\varphi(1+\rho))^q \frac{(\alpha^{r-q})^2}{\alpha^r} \leq C \tag{8}$$

*then for some constant $C > 0$, we have the following result after $R$ federated rounds:*

$$\frac{1}{\sum_{r=1}^{R} \alpha^r} \sum_{r=1}^{R} \alpha^r \mathbb{E}[||\nabla f_i(W_i^r)||^2] \leq \frac{4(f_i(\tilde{W}^0) - f_i(\tilde{W}^*))}{\sum_{r=1}^{R} \alpha^r} + \frac{\frac{2\sigma^2 L}{N}(1 + \frac{2L(1+2\varphi)C}{\rho}) \sum_{r=1}^{R} (\alpha^r)^2}{\sum_{r=1}^{R} \alpha^r}$$

*where $\tilde{W}^*$ is the optimal solution to $f_i$.*

*Proof.* Under the Assumption 3, we have

$$f_i(\tilde{W}^{r+1}) - f_i(\tilde{W}^r) \leq \langle \nabla f_i(\tilde{W}^r), \tilde{W}^{r+1} - \tilde{W}^r \rangle + \frac{L}{2} ||\tilde{W}^{r+1} - \tilde{W}^r||^2$$

$$= -\langle \nabla f_i(\tilde{W}^r), \alpha^r G(W^r) \rangle + \frac{L}{2} ||\alpha^r G(W^r)||^2 \tag{9}$$

and

$$||\nabla f_i(W_i^r)||^2 = ||\nabla f_i(W_i^r) - \nabla f_i(\tilde{W}^r) + \nabla f_i(\tilde{W}^r)||^2$$

$$\leq 2||\nabla f_i(W_i^r) - \nabla f_i(\tilde{W}^r)||^2 + 2||\nabla f_i(\tilde{W}^r)||^2 \tag{10}$$

$$\leq 2L^2 ||W_i^r - \tilde{W}^r||^2 + 2||\nabla f_i(\tilde{W}^r)||^2$$

Taking the expectation at federated round $r$, we bonud

$$\mathbb{E}[f_i(\tilde{W}^{r+1})] - f_i(\tilde{W}^r)$$

$$\leq -\langle \nabla f_i(\tilde{W}^r), \alpha^r \mathbb{E}[G(W^r)] \rangle + \frac{L}{2} \mathbb{E}[||\alpha^r G(W^r)||^2]$$

$$= -\langle \nabla f_i(\tilde{W}^r), \alpha^r \nabla f_i(W^r) \rangle + \frac{L}{2} \mathbb{E}[||\alpha^r G(W^r)||^2]$$

$$= -\frac{\alpha^r}{2} (||\nabla f_i(\tilde{W}^r) - \nabla f_i(W^r)||^2 - ||\nabla f_i(\tilde{W}^r)||^2 - ||\nabla f_i(W^r)||^2) + \frac{(\alpha^r)^2 L}{2} \mathbb{E}[||G(W^r)||^2]$$

$$\leq -\frac{\alpha^r}{2} ||\nabla f_i(\tilde{W}^r)||^2 + \frac{\alpha^r L^2}{2} ||W_i^r - \tilde{W}^r||^2 + \frac{(\alpha^r)^2 L}{2} \mathbb{E}[||G(W^r)||^2]$$

$$\leq -\frac{\alpha^r}{2} (||\nabla f_i(\tilde{W}^r)||^2 + L^2 ||W_i^r - \tilde{W}^r||^2) + \alpha^r L^2 ||W_i^r - \tilde{W}^r||^2 + \frac{(\alpha^r)^2 L\sigma^2}{2N}$$

Taking the expectation before $r$, it yields

$$\mathbb{E}[f_i(\tilde{W}^{r+1})] - \mathbb{E}[f_i(\tilde{W}^r)]$$

$$\leq -\frac{\alpha^r}{2}(\mathbb{E}[||\nabla f_i(\tilde{W}^r)||^2 + L^2||W_i^r - \tilde{W}^r||^2]) + \alpha^r L^2 \mathbb{E}[||W_i^r - \tilde{W}^r||^2] + \frac{(\alpha^r)^2 L\sigma^2}{2N}$$

$$\leq -\frac{\alpha^r}{2}(\mathbb{E}[||\nabla f_i(\tilde{W}^r)||^2 + L^2||W_i^r - \tilde{W}^r||^2])$$

$$+ \frac{(\alpha^r)^2 L\sigma^2}{2N} + \frac{\alpha^r L^2(1+2\varphi)}{\rho}\sum_{q=1}^r (\varphi(1+\rho))^q(\alpha^{r-q})^2\frac{\sigma^2}{N}$$

$$= -\frac{\alpha^r}{2}(\mathbb{E}[||\nabla f_i(\tilde{W}^r)||^2 + L^2||W_i^r - \tilde{W}^r||^2])$$

$$+ \frac{(\alpha^r)^2 L\sigma^2}{2N} + \frac{(\alpha^r)^2 L^2(1+2\varphi)}{\rho}\sum_{q=1}^r (\varphi(1+\rho))^q\frac{(\alpha^{r-q})^2}{\alpha^r}\frac{\sigma^2}{N}$$

Apply (8) to the above inequality, we get

$$\mathbb{E}[f_i(\tilde{W}^{r+1})] - \mathbb{E}[f_i(\tilde{W}^r)]$$

$$\leq -\frac{\alpha^r}{2}(\mathbb{E}[||\nabla f_i(\tilde{W}^r)||^2 + L^2||W_i^r - \tilde{W}^r||^2]) + \frac{(\alpha^r)^2 L\sigma^2}{2N} + \frac{(\alpha^r)^2 L^2(1+2\varphi)C\sigma^2}{\rho N}$$

$$= -\frac{\alpha^r}{2}(\mathbb{E}[||\nabla f_i(\tilde{W}^r)||^2 + L^2||W_i^r - \tilde{W}^r||^2]) + \frac{(\alpha^r)^2 \sigma^2 L}{2N}(1 + \frac{2L(1+2\varphi)C}{\rho})$$

Combine with inequality (10), we can obtain

$$\alpha^r \mathbb{E}[||\nabla f_i(W_i^r)||^2] \leq 2\alpha^r(\mathbb{E}[||\nabla f_i(\tilde{W}^r)||^2 + L^2||W_i^r - \tilde{W}^r||^2])$$

$$\leq 4(\mathbb{E}[f_i(\tilde{W}^{r+1})] - \mathbb{E}[f_i(\tilde{W}^r)]) + \frac{2(\alpha^r)^2\sigma^2 L}{N}(1 + \frac{2L(1+2\varphi)C}{\rho})$$

Summing up the above inequality for $r = 1, 2, \cdots, R$, we have

$$\sum_{r=1}^R \alpha^r \mathbb{E}[||\nabla f_i(W_i^r)||^2] \leq 4(f_i(\tilde{W}^0) - f_i(\tilde{W}^*)) + \frac{2\sigma^2 L}{N}(1 + \frac{2L(1+2\varphi)C}{\rho})\sum_{r=1}^R (\alpha^r)^2 \quad (11)$$

By dividing the summation of learning rates and therefore:

$$\frac{1}{\sum_{r=1}^R \alpha^r}\sum_{r=1}^R \alpha^r \mathbb{E}[||\nabla f_i(W_i^r)||^2] \leq \frac{4(f_i(\tilde{W}^0) - f_i(\tilde{W}^*))}{\sum_{r=1}^R \alpha^r} + \frac{\frac{2\sigma^2 L}{N}(1 + \frac{2L(1+2\varphi)C}{\rho})\sum_{r=1}^R (\alpha^r)^2}{\sum_{r=1}^R \alpha^r}$$

$$(12)$$

The condition (8) keeps right if $\varphi(1+\rho) < 1$. To derive the bound of $\rho$, we have

$$\varphi(1+\rho) = (1 - \frac{k}{n})(1+\rho) < 1$$

Therefore, one should choose $\rho < \frac{k}{n-k}$ to satisfy the above inequality. Theorem 2 implies that each client's model in the FedPSE framework will converge if federated round $R$ is large enough when $\alpha^r$ satisfies the following conditions:

$$\lim_{R\to\infty}\sum_{r=1}^R \alpha^r = \infty \text{ and } \lim_{R\to\infty}\frac{\sum_{r=1}^R (\alpha^r)^2}{\sum_{r=1}^R \alpha^r} = 0$$

$\square$

**Corollary 1.** *Under the assumptions in Theorem 2, if $\tau = \varphi(1+\rho)$ and $\alpha^r = \theta\sqrt{\frac{MN}{R}}, \forall r > 0$, where $\theta > 0$ is a constant, we have the convergence speed of FedPSE:*

$$\mathbb{E}[\frac{1}{R}\sum_{r=1}^R ||\nabla f_i(W_i^r)||^2] \leq \frac{4(f_i(\tilde{W}_i^0) - f_i(\tilde{W}_i^*))}{\theta\sqrt{MNR}} + \frac{2\theta L\sigma^2\sqrt{M}}{\sqrt{NR}} + \frac{4\sigma^2 L^2(1+2\varphi)\tau\theta^2 M}{R\rho(1-\tau)}$$

*Proof.* First we prove that $\alpha^r = \theta\sqrt{\frac{MN}{R}}$, a constant step size, satisfies the inequality (8). we set $\alpha^r = \alpha$ for simplification:

$$\sum_{q=1}^{r}(\varphi(1+\rho))^q\frac{(\alpha^{r-q})^2}{\alpha^r} = \sum_{q=1}^{r}\tau^q\frac{(\alpha^{r-q})^2}{\alpha^r} = \alpha\sum_{q=1}^{r}\tau^q = \alpha\frac{\tau(1-\tau^r)}{1-\tau}$$

Since $0 \le r < 1$, we then obtain

$$\lim_{r\to\infty}\alpha\frac{\tau(1-\tau^r)}{1-\tau} = \frac{\alpha\tau}{1-\tau}$$

Therefore, inequality (8) holds under the condition: $C = \frac{\alpha\tau}{1-\tau}$. From Theorem 2, we obtain the inequality of the expected average-squared gradients of $f_i$, i.e.,

$$\mathbb{E}[\frac{1}{T}\sum_{r=1}^{R}||\nabla f_i(W_i^r)||^2] \le \frac{4(f_i(\tilde{W}_i^0) - f_i(\tilde{W}_i^*))}{\alpha R} + \frac{2\sigma^2 L}{N}(1 + \frac{2L(1+2\varphi)C}{\rho})\alpha$$

$$= \frac{4(f_i(\tilde{W}_i^0) - f_i(\tilde{W}_i^*))}{\theta\sqrt{MNR}} + \frac{2\theta L\sigma^2\sqrt{M}}{\sqrt{NR}} + \frac{4\sigma^2 L^2(1+2\varphi)\tau\theta^2 M}{R\rho(1-\tau)}$$

$$\square$$

From Corollary 1, we can conclude that the framework FedPSE has a convergence rate of $\mathcal{O}(\frac{1}{\sqrt{R}})$ with a proper learning rate,. It also indicates that the hyper-parameter $k$ (in Top-K) has minor impacts on the convergence rate if $R$ is large enough.

## C  DETAILS OF EXPERIMENTS SETTINGS

### C.1  SEPARATION OF CLIENTS' DATASETS

In this section, we propose the details of the splitting method with a "flat" existing dataset in the Non-IID setting. Firstly, the whole training/test samples are separated into two parts, including the heterogeneous subset with $\lambda$ percent and the homogeneous subset with $(1.0 - \lambda)$ percent. On one hand, the homogeneous subset is averaged partitioned into $N$ clients randomly. On the other hand, the heterogeneous subset is sorted by labels and split into $N$ clients by order, which indicates that each client has its own distribution. Finally, we combine the above-mentioned homogeneous part and heterogeneous part into the individual training/test dataset of each client.

### C.2  MODELS FOR DIFFERENT DATASETS

We establish an ConvNet2 model (Beutel et al., 2020) for MNIST and FMNIST datasets (Subramani et al., 2021). In similarity, we construct a model with one Embedding layer, which is prebuilt in Keras, followed by a fully-connected layer on the IMDB dataset. Finally, we train the Resnet18 model (He et al., 2016) on the Cifar10.

### C.3  HYPER-PARAMETERS OPTIMIZATION

We fix the architectures of models and the random seed as $1$. Then we optimize the hyper-parameters(e.g. batch size ranging from $64$ to $512$ and learning rate ranging from $1e-3$ to $1e-2$) for different strategies to compare their best metrics in the following experiments. Particularly, in order to accelerate the training process of Resnet18 on Cifar10, we use a pre-trained model to initialize the clients' weights. The experiments are conducted in a stand-alone PC to simulate the communication in federated learning.

## D  SUPPLEMENTARY EXPERIMENTS

### D.1  PERFORMANCE COMPARISON OF ALGORITHMS WITH DIFFERENT NUMBERS OF CLIENTS

In order to prove the effectiveness of our method, we vary the number of clients from 2 to 100 and report their accuracy in Table 2, where we perform the experiments on the FMNIST and IMDB

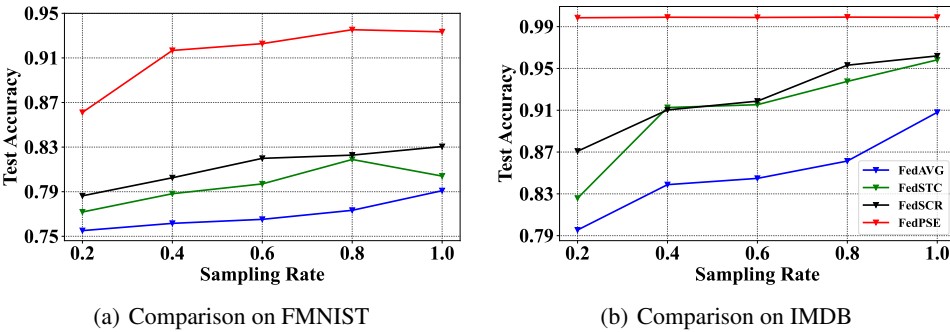

(a) Comparison on FMNIST                    (b) Comparison on IMDB

Figure 7: Accuracy comparison over sampling ratios on Non-IID datasets.

datasets with $\lambda = 1.0$. Then we compute the promotion of our method compared to the SOTA methods via the following function:

$$Promotion = \frac{Acc_{FedPSE} - Acc_{SOTA}}{Acc_{SOTA}}$$

Then we abbreviate the promotion of FedPSE compared to FedSTC as P1 and the promotion of FedPSE compared to FedSCR as P2 as shown in Table 2. From Table 2, we have the conclusion that our proposed FedPSE achieves the best performance among all algorithms under the different numbers of clients, which is consistent with Section 5.2.

Table 2: Performance comparison of algorithms with the different number of clients on Non-IID datasets.

| Datasets | FMNIST | | | | | | IMDB | | | | | |
|---|---|---|---|---|---|---|---|---|---|---|---|---|
| clients | FedAVG | FedSTC | FedSCR | FedPSE | P1 | P2 | FedAVG | FedSTC | FedSCR | FedPSE | P1 | P2 |
| 2 | 0.8781 | 0.8844 | 0.9031 | **0.9466** | 7.03% | 4.82% | 0.8406 | 0.8409 | 0.8438 | **0.9953** | 17.33% | 11.65% |
| 5 | 0.8068 | 0.8756 | 0.8821 | **0.9267** | 5.84% | 5.06% | 0.8494 | 0.9066 | 0.9366 | **0.9966** | 9.78% | 6.55% |
| 10 | 0.7909 | 0.8614 | 0.8603 | **0.9371** | 8.79% | 8.93% | 0.9080 | 0.9582 | 0.9620 | **0.9989** | 4.17% | 3.72% |
| 25 | 0.8045 | 0.8553 | 0.8612 | **0.9021** | 5.47% | 4.75% | 0.8570 | 0.9104 | 0.9112 | **0.9923** | 10.30% | 8.99% |
| 50 | 0.8071 | 0.8343 | 0.8531 | **0.8939** | 7.14% | 4.78% | 0.8356 | 0.9003 | 0.9085 | **0.9986** | 10.92% | 9.92% |
| 100 | 0.7921 | 0.8211 | 0.8344 | **0.8772** | 6.83% | 5.13% | 0.8401 | 0.9036 | 0.9101 | **0.9975** | 10.41% | 9.63% |
| Average | 0.8133 | 0.8553 | 0.8657 | **0.9139** | 6.85% | 5.57% | 0.8551 | 0.9017 | 0.9120 | **0.9965** | 10.68% | 9.43% |

## D.2 PERFORMANCE COMPARISON OF ALGORITHMS IN THE PARTIAL CLIENT PARTICIPATION SCENARIO

We take experiments with partial clients participation for aggregation on the FMNIST and IMDB datasets in the Non-IID setting ($\lambda = 1.0$). Specifically, the clients are randomly sampled by the server at different rates. As shown in Figure 7, our method (red line) significantly outperforms other models, which is align with the conclusion in Section 5.2.

## D.3 VARIANCE OF CORRELATION DISTANCE

Table 3: Variance of correlation distances in FedPSE with different Non-IID ratios.

| Datasets | FMNIST | | | IMDB | | |
|---|---|---|---|---|---|---|
| clients | 2 clients | 5 clients | 10 clients | 2 clients | 5 clients | 10 clients |
| $\lambda = 1.0$ | 5.8e-3 | 6.3e-3 | 2.0e-3 | 2.8e-3 | 1.2e-2 | 1.7e-3 |
| $\lambda = 0.5$ | 3.6e-5 | 4.7e-5 | 4.0e-4 | 2.3e-4 | 4.5e-4 | 1.3e-3 |
| $\lambda = 0.0$ | 1.0e-8 | 3.3e-6 | 2.9e-4 | 1.7e-4 | 2.0e-4 | 4.1e-4 |

In this section, we compute the variance of correlation distances with diverse Non-IID ratios $\lambda$ as shown in Table 3, in which we perform the experiments on the FMNIST and IMDB datasets with a variable clients' number. Then we deduce our conclusion in section 5.4.

### D.4 CONTRIBUTION OF THE DPS ALGORITHM

In order to prove the effectiveness of the DPS algorithm, we construct experiments of FedPSE with different downstream updates on the FM-NIST datasets over Non-IID ratios $\lambda$ (i.e. from 0.0 to 1.0) as shown in Figure 5. The FedPSE without the DPS algorithm indicates that the server sparsifies the global updates with the TopK method and broadcasts the same sparse information to all clients. Apparently, the accuracy of FedPSE with DPS exceeds the algorithm without DPS through different distributions in the Non-IID setting (i.e. $\lambda \geq 0.2$). In conclusion, the DPS method, personalizing the clients' models, is useful for our paradigm with heterogeneous datasets.

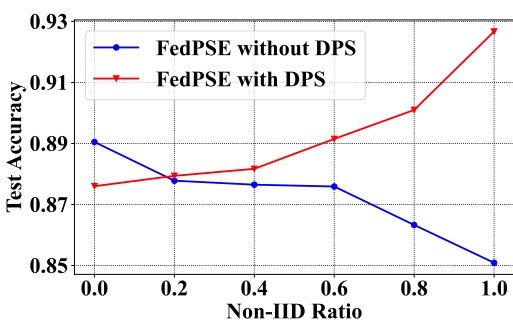

Figure 8: Accuracy with different downstream updates.

## E EXAMPLE CODE

We provide the example code for our framework in this section. Figure 9 shows an example code of the UPS algorithm, while Figure 10 is an example implementation of the EWA algorithm and Figure 12 for the DPS algorithm. Furthermore, we present the training process of FedPSE in Figure 13, in which we assume that there are two clients (a and b) with two-layers neural networks.

```
1   from abc import ABC
2   import numpy as np
3   import tensorflow as tf
4
5
6   # define the client class with the ups algorithm
7   class Client(ABC):
8       def __init__(
9           self, model, ds_train, ds_valid
10      ):
11          self.model = model
12          self.ds_train = ds_train
13          self.ds_valid = ds_valid
14          self.model_weights = []
15          self.res_err = []
16
17      def ups(self, updates, sparsity):
18          # update local model with downstream updates
19          self.model_weights = [
20              np.add(w, u) for w, u in zip(self.model_weights, updates)
21          ]
22          self.model.set_weights(self.model_weights)
23          # train the local model using local dataset
24          hist = self.model.fit(
25              self.ds_train,
26              validation_data=self.ds_valid,
27              epochs=1
28          )
29          n_samples = self.ds_train.cardinality().numpy()
30          # sparsify the upstream updates via topk with error residual
31          self.client_updates = [
32              np.subtract(new_w, old_w)
33              for new_w, old_w in zip(self.model.get_weights(), self.model_weights)
34          ]
35          merged_updates = [
36              np.add(new_u, err)
37              for new_u, err in zip(self.client_updates, self.res_err)
38          ]
39          upstream_updates = topk_algorithm(merged_updates)
40          self.res_err = [
41              np.subtract(lu, su)
42              for lu, su in zip(merged_updates, upstream_updates)
43          ]
44
45          return upstream_updates, n_samples
```

Figure 9: Example code of the UPS algorithm.

```
1   from functools import reduce
2   import numpy as np
3
4
5   # define the function of the EWA algorithm
6   def ewa(clients_updates, clients_samples):
7       # get the participated samples of each element
8       weighted_num = [
9           [np.sign(np.absplute(layer)) * num for layer in updates]
10          for updates, num in zip(clients_updates, clients_samples)
11      ]
12      num_samples = [
13          reduce(np.add, layer_num) for layer_num in zip(*weighted_num)
14      ]
15
16      # get the weighted sum of updates on each index
17      weighted_updates = [
18          [layer * num for layer in updates]
19          for updates, num in zip(clients_updates, clients_samples)]
20      sum_updates = [
21          reduce(np.add, layer_updates) for layer_updates in zip(*weighted_updates)
22      ]
23
24      # get the element-wised updates
25      element_wise_updates = [
26          np.divide(u, n)
27          for u, n in zip(sum_updates, num_samples)
28      ]
29
30      return element_wise_updates
```

Figure 10: Example code of the EWA algorithm.

```
1  import numpy as np
2
3
4  def merge_indices(server_indices, client_indices, dist):
5      merged_indices = []
6      # compute the merged indices of each layer
7      for s_ind, c_ind, d in zip(server_indices, client_indices, dist):
8          # get the intersection of index matrices
9          inter_ind = np.multiply(s_ind, c_ind)
10         i_nonzero_num = np.count_nonzero(inter_ind)
11         inter_index = np.flatnonzero(inter_ind)
12         # compute the corresponding compensation
13         s_nonzero_num = np.count_nonzero(s_ind)
14         server_c = s_ind - inter_ind
15         client_c = c_ind - inter_ind
16         server_c_index = np.flatnonzero(server_c)
17         client_c_index = np.flatnonzero(client_c)
18         # select the indices from the compensations
19         client_num = round(d * (s_nonzero_num - i_nonzero_num))
20         server_num = round((1.0 - d) * (s_nonzero_num - i_nonzero_num))
21         server_index = np.sort(
22             np.random.choice(server_c_index, server_num, replace=False)
23         )
24         client_index = np.sort(
25             np.random.choice(client_c_index, client_num, replace=False)
26         )
27         # get the merged indices
28         merge_index = np.concatenate((inter_index, server_index, client_index))
29         indices_array = np.zeros(s_ind.shape).flatten()
30         indices_array[merge_index] = 1
31         merged_indices.append(indices_array.reshape(s_ind.shape))
32     return merged_indices
```

Figure 11: Example code of the utils.

```
1   import numpy as np
2   from utils import merge_indices
3
4
5   # define the function of the DPS algorithm
6   def dps(c_updates, s_updates, s_sparse_updates):
7       clients_down_updates = []
8       # get the index matrices of the server
9       server_indices = [
10          np.sign(np.absolute(layer))
11          for layer in s_updates
12      ]
13      s_norm = [w / np.linalg.norm(w) for w in s_updates]
14      # get the downstream updates of each client
15      for client_updates in c_updates:
16          # get the index matrices of the client
17          client_indices = [
18              np.sign(np.absolute(layer))
19              for layer in client_updates
20          ]
21          # compute the correlation distance
22          client_norm = [w / np.linalg.norm(w) for w in c_updates]
23          dist = [
24              0.5 - 0.5 * cosine_distance(s_layer, c_layer)
25              for s_layer, c_layer in zip(s_norm, client_norm)
26          ]
27          # get the combined indices
28          merged_indices = merge_indices(
29              server_indices,
30              client_indices,
31              dist
32          )
33          # compute the downstream personalized updates
34          client_down_updates = [
35              np.multiply(layer, index)
36              for layer, index in zip(s_updates, merged_indices)
37          ]
38          clients_down_updates.append(client_down_updates)
39      return tuple(clients_down_updates)
```

Figure 12: Example code of the DPS algorithm.

```
1  import numpy as np
2  import tensorflow as tf
3
4  from ewa import ewa
5  from dps import dps
6  from ups import Client
7
8
9  # define a neural network model
10 def build_model(input_shape):
11     model = tf.keras.Sequential(
12         [
13             tf.keras.Input(shape=input_shape),
14             tf.keras.layers.Dense(50, activation="relu"),
15             tf.keras.layers.Dense(10, activation="relu"),
16         ]
17     )
18     model.compile(
19         optimizer=tf.keras.optimizers.Adam(),
20         loss=tf.keras.losses.categorical_crossentropy,
21         metrics=["accuracy"]
22     )
23     return model
24
25 # main process
26 # initial hyper-parameters
27 sparsity = 0.9
28 max_round = 100
29
30 # clients load data
31 a_ds_train, a_ds_valid, feature_dim = load_tf_datasets("data_a_location")
32 b_ds_train, b_ds_valid, feature_dim = load_tf_datasets("data_b_location")
33
34 # define model
35 model = build_model(feature_dim)
36 client_a = Client(model, a_ds_train, a_ds_valid)
37 client_b = Client(model, b_ds_train, b_ds_valid)
38 a_down_updates = []
39 b_down_updates = []
40 for _ in range(max_round):
41     a_up_updates, a_samples = client_a.ups(a_down_updates, sparsity)
42     b_up_updates, b_samples = client_b.ups(b_down_updates, sparsity)
43     clients_updates = [a_up_updates, b_up_updates]
44     clients_samples = [a_samples, b_samples]
45     s_updates = ewa(clients_updates, clients_samples)
46     s_sparse_updates = topk_algorithm(s_updates)
47
48     a_down_updates, b_down_updates = dps(
49         clients_updates, s_updates, s_sparse_updates
50     )
51
52 client_a.model.save('client_a_model.h5')
53 client_b.model.save('client_b_model.h5')
```

Figure 13: Example code of the main process.

