# OpenReview forum: "FedPSE: Personalized Sparsification with Element-wise Aggregation for Federated Learning"
_ICLR.cc/2023/Conference — Submitted to ICLR 2023_

### Official Review · Reviewer_7ra3 · 2022-10-25

**Confidence:** 3
**Correctness:** 3
**Technical Novelty And Significance:** 2
**Empirical Novelty And Significance:** 2
**Recommendation:** 3

**Clarity, Quality, Novelty And Reproducibility:**

The paper is mostly clear, with a few confusing parts that I mentioned above. The hyperparameters are not provided anywhere (including in the appendix), and the authors didn't share the code for reproducibility.

**Details Of Ethics Concerns:**

I don't think the paper needs an ethics review, but I expressed my fairness concern above. I am happy to revisit it if the authors address my concern.

**Strength And Weaknesses:**

Strengths:
- The paper cares about bidirectional communication cost, which is nice as server-to-client communication cost is mostly neglected in the literature.
- The authors provide theoretical convergence guarantees.

Weaknesses:
- I am confused about the Sign(.) operation in Algorithms 2 and 3. What is the point of taking the sign of a nonnegative matrix $|\Delta \hat{W}^r_i|$?

- The upstream sparsification and element-wise aggregation are part of many existing frameworks. The only novel block seems to be the downstream personalized sparsification.

- In Section 2.3, it says FedSTC and FedSCR share the uniform model. What does that mean? What I understand from a uniform model is a model with uniformly distributed parameters. But I think the authors meant that each client has the same model. I think changing the terminology here would avoid some potential confusion about the prior work.

- Since each client have a different dataset distribution and a task, wouldn't it make more sense to choose different sparsity ratios for each client? By forcing the same sparsity ratio on all clients, FedPSE performs worse for clients whose datasets require denser models. So I think this approach brings a fairness problem. Do the authors have a solution for this?

- In Section 5.1, the authors said they used the same learning rate for all compression methods. The learning rate and most other hyperparameters should be tuned for each baseline for the best performance. Otherwise, the comparisons are not valid and they would favor the method the authors tuned the learning rate for.

- It would make reading easier if the references for FedAVG, FedSTC, and FedSCR were included in the Table captions, or at least somewhere in the experiments section.

- What is the strategy to communicate the sparse models between the server and the clients? Once the model update is sparsified, it should be encoded in a way that the sparsity helps reduce the bitrate. What is the proposed strategy there?

- It seems from Table 1 that FedPSE is losing the performance gain for $\lambda < 1$? While it outperforms the baselines consistently for $\lambda=1$, it either underperforms or performs almost the same as the baselines for $\lambda < 1$. Does this imply that FedPSE does not provide much advantage for not purely heterogeneous datasets? A comment on this in the experiments section would be nice.

**Summary Of The Paper:**

The authors propose a personalized sparsification strategy for federated learning that compresses the updates in both directions, i.e., from clients to server and from server to clients. The proposed method, FedPSE, achieves that by (1) sparsification of model updates using Top-k algorithm before communication from clients to server, (2) element-wise aggregation at the server, and (3) sparsifying the global model updates in a personalized way before communication from the server to the clients. The experimental results show that FedPSE outperforms FedAVG consistently, and FedSTC and FedSCR occasionally on MNIST and FMNIST datasets.

**Summary Of The Review:**

Please see my comments above.

---

> ### Author Response · Authors · 2022-11-17
> **Thank you for your valuable comments! Reply to Reviewer 7ra3.**
>
> We thank the reviewer for the positive and constructive comments. We provide our responses as follows.
>
> 1. **Q**: What is the point of the Sign$(\left\| \cdot \right\|)$ function in Algorithms 2 and 3?
>
> **A**: The purpose of the Sign$(\left\| \cdot \right\|)$ function is to generate the index matrix of inputs. To be specific, the non-zero elements of inputs are set to 1, while the zero elements are unchangeable. Then the index matrix is used by the following functions as shown in Algorithms 2 and 3. We have added a detailed description of the Sign$(\left\| \cdot \right\|)$ function in the latest version.
>
>
> 2. **Q**: The upstream sparsification and element-wise aggregation are part of many existing frameworks. The only novel block seems to be the downstream personalized sparsification.
>
> **A**: As we have claimed in Section 1.3, our contribution includes three blocks: the overall efficient framework with personalization concern (FedPSE), the element-wise aggregation method (EWA), and the downstream personalized sparsification~(DPS).
> Firstly, our framework outperforms the SOTA methods in the same setting, which has been proved by the experiments in Table 1.
> Secondly, to the best of our knowledge, we are the first ones to utilize the EWA method to aggregate the sparse matrices. The current works compute the aggregation by model-wised averaging \cite{} or layer-wised aggregating \cite{}. Can you supply more information about the existing frameworks?
> Thirdly, in order to prove the effectiveness of UPS, we take more experiments on FedPSE without DPS block, please see more details in Appendix D.4.
>
> 3. **Q**: The terminology of the uniform model should be replaced by the same model.
>
> **A**:  We have fixed this misleading terminology in our latest version.
>
> 4. **Q**: It would make more sense to choose different sparsity ratios for each client.
>
> **A**: Thanks for your advice. We will optimize the sparsity ratios for each client adaptively and dynamically in our future work.
>
> 5. **Q**: The learning rate and most other hyperparameters should be tuned for each baseline for the best performance.
>
> **A**: We have tuned the hyperparameters for different strategies and updated the results in our latest version.
>
> 6. **Q**: The references for FedAVG, FedSTC, and FedSCR should be included in the Table captions.
>
> **A**: Due to the limitation of the table space, we only fix the references in the comments of experiments as shown in Section 5.2.  Please see more details in our new version.
>
> 7. **Q**: the strategy that communicates the sparse models between the server and the clients.
>
> **A**: The experiments are conducted in a stand-alone PC to simulate the communication cost in federated learning.
> We can leverage the sparse.csr_matrix API in the scipy[1] package to compress the sparse matrices between the server and the clients in the real-world setting, which has a higher compression ratio with a faster computation rate.
>
> 8. **Q** more comments on Table 1 should be proposed.
>
> **A**: Our proposal aims to resolve the bidirectional communication challenge in the Non-IID setting the same as current works, i.e. FedSTC and FedSCR, which perform not well enough on the heterogenous datasets.
> We have updated our results after the hyper-parameters optimization which has the conclusion that our proposed FedPSE almost achieves the best performance of all algorithms in the Non-IID setting ($\lambda=1.0 or 0.5$).
> Please see more details in Section 5.2 in our latest paper.
>
> 9. **Q** The hyperparameters are not provided anywhere, and the authors didn't share the code for reproducibility.
>
> **A**: We have added the hyperparameters and example code of FedPSE in Section C.3 and Appendix E, please see more details in the latest version.
>
> [1] Virtanen P, Gommers R, Oliphant T E, et al. SciPy 1.0: fundamental algorithms for scientific computing in Python[J]. Nature methods, 2020, 17(3): 261-272.

---

### Official Review · Reviewer_DP2J · 2022-10-30

**Confidence:** 3
**Correctness:** 3
**Technical Novelty And Significance:** 2
**Empirical Novelty And Significance:** 2
**Recommendation:** 6

**Clarity, Quality, Novelty And Reproducibility:**

Clarity: the paper is lack of a full screen that contain details of the whole method. It would be better to include pseudocode.

Quality: The motivation of the paper is good.

Reproducibility: It seems not straightforward to reproduce the proposed algorithm based on the materials provided so far.

**Strength And Weaknesses:**

1. This paper is not the first one to utilize element-wise aggregation in FL to enhance the performance in the non-iid setting. For example, pFedLA (Layer-wised Model Aggregation for Personalized Federated Learning), which was published in CVPR 2022, also utilizes element-wise aggregation in FL. So the comparison of the proposed method and pFedLA is required to be included in the paper's analysis as well as experiments.

2. The details of the experiments are missed, such as learning rate, random seed, batch size.

3. The experiments have not show the advantage of the communication efficiency. It only compare with the algorithms that also have improve the communication efficiency. However, it's necessary to compare with the SOTA personalized federated learning approach that didn't consider the communication efficiency, to show the necessity of communication efficiency. For example, if your approach performs much worse than those general personalized FL, then the FL operators may insist to use general SOTA personalized FL. Besides, it's necessary to do a experiment to compare with a method that skipping the communication efficiency enhanced strategy of your proposed algorithm to show the contribution to reducing the communication cost; on the other hand, it's necessary to compare with that skipping the
element-wise aggregation to show it's useful to set up such a component.

**Summary Of The Paper:**

This paper focuses on enhance the performance of non-iid FL on the basis of reducing the communication cost.
The author proposes a method that uses the personalized Top-K sparsification and element-wise average techniques.
The main contribution of the paper is that it proposes a downstream selection mechanism to personalize the clients' models, which adapts to various distributions and can increase the performance in the non-iid setting.

**Summary Of The Review:**

In short, the motivation of the paper is good to me. The author(s) may consider doing more experiments as mentioned above to show the strengths of the proposed algorithm. Besides, it would be more convincing if pseudocode/code is provided.

---

> ### Author Response · Authors · 2022-11-17
> **Thank you for your valuable comments! Reply to Reviewer DP2J.**
>
> We thank the reviewer for the positive and constructive comments. We provide our responses as follows.
>
> 1. **Q**: As pFedLA is the first one to utilize element-wise aggregation, the comparison should be included in the paper's analysis as well as experiments.
>
> **A**: To the best of our knowledge, we are the first ones to utilize the element-wised aggregation method, which is suitable for sparse aggregated matrices.
> The layer-wised aggregation process in pFedLA is different from the element-wised aggregation method in our proposal.
> Firstly, pFedLA aims to personalize the clients' models to promote performance in the Non-IID setting, which ignores the communication challenge in FL.
> Secondly, the elements in the same layer share the uniform weight to aggregate the global layer in pFedLA, while each element of the same layer has its individual weight for aggregation in our method.
> Thirdly, pFedLA aggregates the dense matrices from participated clients, while our algorithm averages the sparse matrices with a high compression ratio.
> Nevertheless, we have surveyed the mentioned papers and taken them into consideration in our latest version (see Section 2.2).
>
> 2. **Q**: The details of the experiments are missed, such as learning rate, random seed, and batch size.
>
> **A**: Thanks for your comments. We have added the mentioned hyper-parameters in Appendix C.3, please see more details in the latest version.
>
> 3. **Q**: It's necessary to compare with the SOTA personalized FL approach without communication compression. It's necessary to compare with a method that skipping the communication efficiency enhanced strategy of your proposed algorithm and compare with that skipping the element-wise aggregation.
>
> **A**: Our proposal aims to reduce the bidirectional communication cost with acceptable performance in the Non-IID setting.
> Firstly, communication efficiency is one of the most important challenges in Federated Learning, which has massive information transmission in the distributed scenario.
> As a result, it is not necessary to compare our method with the SOTA personalized FL approaches without communication compression.
> Even though, we have researched the SOTA personalized FL works in Section 2.
>
> Secondly, we have compared FedPSE with the method that skipping the communication efficiency enhanced strategy.
> Apparently, the FedPSE framework is equal to the general FedAVG, when the communication compression is ignored. We have shown the performance comparison in Table 1, in which the results demonstrate the advantage of our proposal.
>
> Thirdly, we have compared the method that skipping the element-wise aggregation as shown in Section 5.5, which demonstrates the effectiveness of our proposed EWA method.
>
> 4. **Q**: It would be better to include pseudocode or example code.
>
> **A**:  We have added the example code of FedPSE in Appendix E, please see more details in the latest version.

---

### Official Review · Reviewer_E2iq · 2022-10-30

**Confidence:** 4
**Correctness:** 3
**Technical Novelty And Significance:** 3
**Empirical Novelty And Significance:** 3
**Recommendation:** 5

**Clarity, Quality, Novelty And Reproducibility:**

Overall, the paper is well-written. But I do think the motivation of this paper is not clear and needs to be revised, as mentioned in the weakness section.

**Strength And Weaknesses:**

Strength
1. The proposed element-wise aggregation method seems to be very promising. It can significantly increase the performance of compression algorithms in FL.
2. Combining all new techniques, the proposed algorithm can achieve significant improves over existing algorithms, especially on CIFAR-10 dataset.


Weakness
1. Weak motivation: The first key problem I found is that the problem this paper aims to address may not even exist. In the introduction, the authors claimed that "the current compression techniques, ignoring the personalization of clients, face a significant performance degradation on Non-IID datasets" However, if we look at table 1, previous FedSTC and FedSCR algorithms seems to be even more robust to the data heterogeneity than FedAvg. In many cases, the performance of FedSCR and FedSTC even increases along with the increase of data heterogeneity. These experimental observations make the motivation of this paper invalid. The authors may need significant efforts to change the wording in the whole paper.
2. Misleading name: The name of "element-wise aggregation" is confusing. Because in FedAvg and many other basic FL algorithms, we also conduct element-wise averaging over clients. What the authors really does here is element-wise normalization. I suggest the authors to change the method name.
3. Compared with previous communication compression works, I think one of the key differences in this paper is downstream personalization. Clients will download different parts of the global model to begin the next round. So a key question here is that how effective is this technique? The authors did not provide ablation study on this. It is possible that this technique is unnecessarily complex. For example, how about transferring the same compressed global model to all clients?
4. The theoretical analysis is a bit weak. As there is no local updates at clients, which is the basic component in FL algorithms. Also, it is unclear what the exact algorithm the authors analyzed. They remove the local updates part. So are there any components removed from the algorithm in order to conduct the analysis? It would be nice if the authors can write down the exact update rule of the proposed algorithm.

**Summary Of The Paper:**

This paper proposes a new communication compression method for federated learning with heterogeneous data. In particular, the compression scheme is personalized to each client. The authors show that several new techniques, such as downstream compression personalization, and element-wise aggregation, are critical to achieve a good performance in the presence of non-IID data.

**Summary Of The Review:**

Currently I am leaning towards a reject. The main reasons are (1) the motivation of this paper is in conflict with their own experimental observations. So it requires some rewriting; (2) The effectiveness of DPS is unclear. More ablation studies are needed.

---

> ### Author Response · Authors · 2022-11-17
> **Thank you for your valuable comments! Reply to Reviewer E2iq.**
>
> We thank the reviewer for the positive and constructive comments. We provide our responses as follows.
>
> 1. **Q**: Weak motivation: FedSTC and FedSCR algorithms seem to be even more robust to the data heterogeneity than FedAvg and increase along with the increase of data heterogeneity.
>
> **A**: Our proposal aims to resolve the bidirectional communication challenge in the Non-IID setting the same as current works, i.e. FedSTC and FedSCR, which perform not well enough on the heterogenous datasets.
>
> Firstly, the naive FedAVG is too easily biased to act as the baseline of federated learning with Non-IID datasets.
> Secondly, the performance of FedSTC and FedSCR frameworks reduces with the increase of data heterogeneity~($\lambda$), while our method is increasing apparently as shown in Table 1 and Figure 5. Furthermore, our method outperforms FedSTC and FedSCR by about 10 percent with the same compression ratio in the Non-IID setting ($\lambda = 1.0$) as shown in Table 2 of Appendix D.1.
>
> 2. **Q**: Misleading name: The name of "element-wise aggregation" is confusing. FedAvg and many other basic FL algorithms also conduct element-wise averaging over clients. What the authors really does here is element-wise normalization.
>
> **A**: Our method aggregates the global updates via element-wised averaging, while the basic FL algorithms use the model-wised method.
>
> To be specific, the element-wised method has different clients' weights on each index due to the sparsity, while the model-wised averaging shares the same weights on each element. Moreover, we have compared the difference between our proposed EWA with the general model-wised method in Section 4.3 and Appendix A.
>
> Furthermore, the name of our aggregation method prefers the element-wised averaging to the element-wised normlization.
> As the EWA method computes the weighted averaging result on each index, the element-wised normlization aims to adjust the element value on different scales to a notionally common scale or a normal distribution.
>
> 3. **Q**: How to prove the effectiveness of downstream personalization?
>
> **A**: The downstream personalization sparsification~(DPS) contributes to the personalized model of different clients, which promotes the performance significantly in the Non-IID setting.
> To prove the effectiveness of DPS, we construct more experiments to compare the performance of our method with or without DPS.
> Due to the limitation pages, we present the detailed results in Appendix D.4, which is updated in the newly updated version.
>
> 4. **Q**: As there are no local updates at clients, the theoretical analysis is a bit weak.
>
> **A**: We have presented the overview of our framework in Section 4.5 and Algorithm 4, in which the UPS block includes the local updates process at clients.
> As shown in the UPS Algorithm, the clients' models are initialized by the received updates from the server and updated by the optimizer with local data as shown in Algorithm 1 from line 3 to line 8.
> Furthermore, we prove the convergence of PedPSE theoretically in Appendix B, which has considered the local updates as shown in Equation 1 of Appendix B.1.

---

### Author Response · Authors · 2022-11-17
**General response**

We thank the reviewers for their valuable comments. We are glad that the reviewers found that we proposed a very promising federated learning framework with efficiency and performance concerns (Reviewers E2iq, DP2J). Both Reviewer DP2J and Reviewer 7ra3 thought that "the motivation is good" and the compression on the "bidirectional communication cost was nice"; Reviewer E2iq believed that our method "achieved significant improves over existing algorithms"; Reviewer 7ra3 believed that our convergence guarantee was proved clearly. We thank all the reviewers’ feedback on the presentation, and we have accordingly modified the paper in the newly updated version.

We appreciate all the feedback and try our best to address most if not all concerns raised by the reviewers, it will be great if the scores can be kindly reconsidered.
In the following, we respond to the comments of each reviewer.

---

### Public Comment · ~Annette_Cheng1 · 2022-11-19
**To reviewers and authors, regarding some critical issues associated with the current manuscript.**

Dear authors and reviewers,
I really enjoyed reading the paper and understanding the scope of the work, particularly from the reviewer’s comments and the authors' rebuttal. In this context I would like to highlight the following points that I think may further help the evaluation of the current manuscript.

**Important related works on recent development in literature of FL that addresses the issue highlighted**

*I understand Section 4.1 clearly mentioning the goal of the work is to be communication efficient. I believe there has been quite a lot of work that the authors might have missed, including (A-D), that not only addressed the issue of communication load but also computation overhead (B-D).*

**Concern with EWA as a contribution: FedDST (D) presented the idea of *"sparse weighted averaging"* that is same as that of EWA (which is claimed to be a contribution of the current manuscript)**

*I would like to take this opportunity to route the readers to the fact that the idea of element wise averaging for sparse models was introduced in FedDST(D) [equation 3 in the Methodology section] and the authors of that manuscript have used this form of averaging for the same purpose of "normalizing the weights based on their participation rate", as the current manuscript.*

*Please note: FedDST(D) also demonstrated the application of this in non-iid dataset (pathologically non-iid to be precise).*

**Concern with UPS as a part of the contribution: Both (B,D) successfully demonstrated various variants of top-K sampling similar (can be argued to be even better) to the what is done in UPS**

*I would like to highlight that the idea of top-K sampling is not new (as partly mentioned by the authors as well). In particular, few recently published works  (e.g.: (D) in AAAI 2022, (B) in ICLR 2022) have taken this idea one step further to demonstrate top-K sampling while sparsely updating the weights locally. This ensures not only communication benefits, but also computation benefits (D). (B) even comprehensively demonstrated various variants of this top-K sampling.*

**FedPSE has no computation benefits whereas some of the published works has (B,D)**

*As in FedPSE the clients locally update all the weights, and only after local training update top-K, it provides no computation benefits, despite the sparsity choice of 0.9. On the contrary existing literature can have both computation benefits as well (B,D).*

**Concern with utility of different aspects of DPS: FedPSE suffers from significant accuracy drop at moderate sparsity of 0.9 (the plot with x value of 0.1 in Fig. 4)**

*This is significantly concerning and thus further ablation of the real utility of various steps described in DPS is needed. Nevertheless, as already stated existing alternatives demonstrated benefits with sparsity of 0.2 and 0.1 while showing both computation and communication benefits.*

*Moreover, it is well known that magnitude pruning (like top-K in the client side) requires retraining to gain the accuracy. Thus this can also be a potential reason for the authors to get significantly poor accuracy at high sparsity (Fig. 4).*

**Please refer to (A) if the concern is only communication cost**

*As the current work focuses on the communication benefits, I would like to refer the reader to (A) as well. This work showed that sending only sparse masks was sufficient enough for the federated learning paradigm, thus saving 32x communication cost.*

I hope this summary would help the reviewers and authors to assess the contributions of the current manuscript. I have added the references at the bottom for their kind consideration and checking.

(A) Sparse Random Networks for Communication-Efficient Federated Learning, arxiv 2022.
(B) ZeroFL: Efficient On-Device Training for Federated Learning with Local Sparsity, ICLR 2022.
(C) Model Pruning Enables Efficient Federated Learning on Edge Devices, TNNLS 2022.
(D) Federated Dynamic Sparse Training: Computing Less, Communicating Less, Yet Learning Better, AAAI 2022.

---

> ### Author Response · Authors · 2022-11-19
> **Thank you for your valuable comments! Reply to Annette Cheng**
>
> Thanks for the constructive comments. We provide our responses as follows.
> 1. **Q**: Important related works on recent development in literature of FL that addresses the issue highlighted.
>
> **A**: Our motivation of this proposal is to solve the communication challenge with Non-IID datasets in federated learning.
> It's necessary to keep the personalization of clients' models as shown in Section 2.3.
> The works mentioned by the comment focus on communication and computation challenges without personalization concerns.
> Furthermore, we will survey these mentioned papers and take them into consideration in our future version.
>
> 2. **Q**: Concern with EWA as a contribution
>
> **A**: The aggregation process of FedDST needs local model weights and their mask arrays which are uploaded from clients.
> The EWA method only receives sparse updates for averaging which decreases the communication cost significantly.
>
> 3. **Q**: Concern with UPS as a part of the contribution
>
> **A**: As we have claimed in Section 1.3, our contribution includes three blocks: the overall efficient framework with personalization concern (FedPSE), the element-wise aggregation method (EWA), and the downstream personalized sparsification (DPS).
> We leverage the current method, topk sparsification with residual error, to compress the upstream information as shown in Section 4.1.
>
> 4.**Q**: FedPSE has no computation benefits whereas some of the published works has.
>
> **A**: Firstly, our method aims to resolve the communication challenge in the Non-IID setting.
>
> Secondly, there are enough computation resources in the cross-silo FL scenarios. Frankly speaking, our computational cost is slightly higher than FedAVG due to the DPS Algorithm. But the server is enough for the DPS process without training the machine learning model.
>
> Last but not least, the local training method of SOTA works can be friendly applied in our UPS algorithm.
>
> 5. **Q**: Concern with the utility of different aspects of DPS. FedPSE suffers from a significant accuracy drop at moderate sparsity of 0.9. The author gets significantly poor accuracy at high sparsity.
>
> **A**: Firstly, FedPSE performs better with a higher sparsity in the Non-IID setting, which "kills two birds with one stone".
> Apparently, the FedPSE framework with a small sparsity is close to the naive FedAVG.
> It's obvious that FedAVG without personalization is not suitable for the Non-IID datasets.
>
> Secondly, our proposal outperforms the FedSTC and FedSCR significantly with a high sparsity(0.9) as shown in Table 2 of Appendix D.1.
>
> 6. **Q**: Please refer to (A) if the concern is only communication cost.
>
> **A**: Our motivation of this proposal is to solve the communication challenge with Non-IID datasets in federated learning.
> Furthermore, we will survey these mentioned papers (A) and take them into consideration in our future version.

---

### Decision · Program_Chairs · 2023-01-20

**Decision:**

Reject

**Justification For Why Not Higher Score:**

The weaknesses clearly outweigh the strengths of this work.

**Justification For Why Not Lower Score:**

N/A

**Metareview: Summary, Strengths And Weaknesses:**

This paper proposes a new approach to reducing communication overhead in the context of personalized federated learning (FL). The approach combines two key ideas: personalized sparsification and element-wise aggregation. In particular, clients download personalized sparse updates. The paper includes theoretical convergence guarantees in the smooth non-convex setting, and experiments on several datasets commonly used in the FL research community. The experimental results are promising, illustrating benefits in terms of accuracy and reduced communication overhead.

Although there is interest in the proposed approach and the experimental results are promising, several concerns were raised about the submission:
* Several details about the experimental setup were not included in the original submission; this was corrected to some extent during the rebuttal.
* Addition comparison with other personalized FL methods in terms of performance/accuracy should be included.
* During the rebuttal, the authors replied that their theory analyzes the case with local updates, but it is not evident where this is covered in Appendix B. (One reviewer pointed this out, and I independently verified this by carefully checking the appendix.)

In addition to the above, the paper could be strengthened by including additional ablations to better highlight the importance of each component introduced in this work, to better support all of the claims made.

There is also the unsolicited feedback provided by Annette Cheng, raising several questions regarding novelty and the relationship to previous work. I read both the feedback and the author response. Although I believe the same decision would have been reached regardless of this feedback, there are certain responses (e.g., regarding comparison to other work on non-personalized communication-efficient FL) which the authors could have better addressed. I encourage the authors to also seriously consider these suggestions when preparing future versions of this work for submission.

**Summary Of Ac-Reviewer Meeting:**

n/a